EMBO
Molecular Medicine

# Luteolin detoxifies DEHP and prevents liver injury by degrading Uroc1 protein in mice

Huiting Wang[1], Ziting Zhao[1], Mingming Song[1], Wenxiang Zhang [1]✉, Chang Liu [1,2,3]✉ & Siyu Chen [1]✉

## Abstract

**Di-(2-ethylhexyl) phthalate (DEHP), an environmental pollutant, has been widely detected in both environmental and clinical samples, representing a serious threat to the homeostasis of the endocrine system. The accumulation of DEHP is notably pronounced in the liver and can lead to liver damage. The lack of effective high-throughput screening system retards the discovery of such drugs that can specifically target and eliminate the detrimental impact of DEHP. Here, by developing a Cy5-modified single-strand DNA-aptamer-based approach targeting DEHP, we have identified luteolin as a potential drug, which showcasing robust efficacy in detoxifying the DEHP by facilitating the expulsion of DEHP in both mouse primary hepatocytes and livers. Mechanistically, luteolin enhances the protein degradation of hepatic urocanate hydratase 1 (Uroc1) by targeting its Ala270 and Val272 sites. More importantly, *trans*-urocanic acid (*trans*-UCA), as the substrate of Uroc1, possesses properties similar to luteolin by regulating the lysosomal exocytosis through the inhibition of the ERK1/2 signal cascade. In summary, luteolin serves as a potent therapeutic agent in efficiently detoxifying DEHP in the liver by regulating the UCA/Uroc1 axis.**

**Keywords** Visual DEHP Tracing System; Luteolin; Uroc1; Liver Injury
**Subject Categories** Digestive System; Evolution & Ecology; Pharmacology & Drug Discovery

See also: F Cappelli & A Mengozzi

## Introduction

Phthalic acid esters (PAEs) currently hold an unparalleled degree of supremacy in the global plasticizer market, encompassing a commanding market share of ~65% (Li et al, 2023; Zhang et al, 2022). Within this realm, the detrimental ramifications of di(2-ethylhexyl) phthalate (DEHP) on the holistic welfare and vitality of both human beings and the animal kingdom ascend to an insurmountable level of gravity and significance (Genuis et al, 2012). Given the absence of a chemical bond with polymers, DEHP possesses a remarkable propensity to be effortlessly liberated from plastic materials and subsequently permeate the atmosphere, edible substances, or bodily fluids (Mengozzi et al, 2021). As an exogenous reagent, the accumulation of DEHP exposure is notably pronounced in the liver, surpassing its accumulation in other tissues such as the gastrointestinal tract and kidneys (Li et al, 2023), emphasizing the liver as the primary target organ for DEHP exposure. More importantly, emerging epidemiological evidence has highlighted the intrinsic connection between DEHP exposure and an augmented susceptibility to liver injury, and nonalcoholic fatty liver disease (Ding et al, 2021; Xu et al, 2010). This association poses a great threat to public health and the overall stability of social economies.

As a nonpersistent pollutant, DEHP swiftly disseminates throughout peripheral tissues before being efficiently eliminated from the body (Ferguson et al, 2014; Mengozzi et al, 2021). Notably, upon reaching the liver, it undergoes metabolic transformation into even more pernicious compounds compared to its parent chemicals, thereby exerting a pleiotropic effect by disrupting numerous metabolic pathways dominantly by increasing oxidative stress and triggering subclinical inflammation (Mengozzi et al, 2021). Accordingly, numerous monomeric drugs derived from traditional Chinese medicine (TCM), such as lycopene and icariin, renowned for their antioxidative or anti-inflammatory characteristics, have been formulated to alleviate the damaging consequences of DEHP in the liver and other peripheral tissues (Genuis et al, 2012). However, these medications merely provide symptomatic relief without tackling the underlying issue (DEHP itself). Thus, removing the unexpected DEHP accumulation in the liver, while reducing its retention time, may provide a novel therapeutic strategy to eliminate the negative impact of DEHP on human health.

In China, classical formulas in TCM, known as traditional empirical and classic prescriptions recorded in ancient medical books, have accumulated centuries of clinical practice by ancient doctors. Moreover, ancient TCM texts contain numerous records and corresponding treatment methods for detoxification and elimination of toxins. For example, based on the esteemed Compendium of Materia Medica, the "Three Bean Drink" comprising mung beans, adzuki beans, black beans, and licorice exhibits remarkable properties in terms of heat expulsion, detoxification, hepatic nourishment, and pulmonary hydration

[1]State Key Laboratory of Natural Medicines and School of Life Science and Technology, China Pharmaceutical University, Nanjing 211198, China. [2]Jiangsu Provincial University Key Laboratory of Drug Discovery for Metabolic Inflammatory Diseases, China Pharmaceutical University, Nanjing 211198, China. [3]Department of Endocrinology, Nanjing Drum Tower Hospital, China Pharmaceutical University, Nanjing 211198, China. ✉E-mail: wenxiangzhang@cpu.edu.cn; changliu@cpu.edu.cn; siyuchen@cpu.edu.cn

(Yi et al, 2005). With the development of network pharmacology, various chemical monomers have been demonstrated to be dominant and functional ingredients in the treatment of various diseases (Li, 2007). Hence, conducting a comprehensive network pharmacology analysis of detoxification prescriptions in TCM will facilitate the identification of effective drugs that can efficiently detoxify the DEHP residues in the human liver.

Here, we designed a Cy5-labeled single-strand DNA-aptamer-based approach that targets DEHP to establish a visual DEHP tracing system. Meanwhile, we summarized 128 detoxification prescriptions in TCM and performed a net pharmacological analysis. Taking advantage of both the system and net pharmacology, we finally identified luteolin as a potential candidate drug. It is worth noting that luteolin is a highly acclaimed flavonoid renowned for its multifaceted pharmacological effects, particularly its impressive anti-inflammatory, antibacterial, antitumor, antioxidant, and antiviral properties (Franza et al, 2021; Gendrisch et al, 2021a; Kawanishi et al, 2005; Kotanidou et al, 2002). However, the beneficial role of luteolin in detoxifying DEHP in the mouse liver and its direct molecular target remain elusive. Mechanistically, luteolin displays remarkable efficacy in detoxifying excessive DEHP in mouse primary hepatocytes (PHs) and livers by enhancing the degradation of hepatic urocanate hydratase 1 (Uroc1) through targeted binding to the Ala270 and Val272 sites. In addition, *trans*-urocanic acid (*trans*-UCA), as the substrate of Uroc1, exhibits similar properties to luteolin by inhibiting the ERK1/2 signal cascade and the related lysosomal exocytosis. Our findings provide compelling evidence that Uroc1 represents a promising therapeutic target for effectively detoxifying DEHP and its associated pollutants in the liver. Notably, luteolin and its related endogenous metabolite *trans*-UCA exhibit potential as functional drugs and metabolites, respectively, in this process.

# Results

## Network pharmacology analysis for the novel drugs possessing the abilities for the hepatic removal of excess DEHP

To discover novel drugs for the detoxification of DEHP from the liver, we subjected a comprehensive examination to 128 detoxification formulas hailing from TCM derived from the TCM prescriptions (TCMP) and chemical component library of chinese medicine databases (CCLCM) (https://organchem.csdb.cn/scdb/main/tcm_introduce.asp) (Dataset EV1). The specific screening criteria and process are presented in Fig. 1A. We set the criteria for screening candidate compounds as follows: oral availability (OB) ≥ 30%, drug likeness (DL) ≥ 0.18, and half-life ≥4 h. Following these criteria, a total of 21 Chinese herbs were filtered according to an appearance frequency ≥10 times (Appendix Fig. S1A and Appendix Table S1), while 387 effective small molecule compounds were achieved by Traditional Chinese Medicine Systems Pharmacology Database and Analysis Platform (TCMSP) systems analysis (Dataset EV2). To identify the functional monomers, venn and topological analyses were performed on TCM and common effective chemical components, and candidate drugs decreasing DEHP accumulation were identified by screening for effective components with a degree of ≥3. As shown in Fig. 1B, a cluster of 37 monomers was identified, while 14 monomers with a degree number ≥3 were selected

(Appendix Table S2). Furthermore, we performed a CCK-8 assay to evaluate the potential toxicity of these drugs and determine the safe concentration for mouse PHs (Fig. EV1A–L). As shown in Fig. EV1H–J, kaempferol, paeoniflorin and berberine hydrochloride exhibited cytotoxicity at all tested doses. Hence, these three monomers were excluded from further consideration, and the remaining 7 monomers were ultimately chosen as potential drugs that could promote the detoxification of excess DEHP in mouse PHs (Fig. EV1A–G,K,L).

## A Cy5-labeled DNA-aptamer-based strategy for DEHP visualization and drug screening in vitro

The elusive nature of DEHP presents challenges in its detection, leading to a lack of efficient high-throughput screening techniques. In order to achieve DEHP visualization and real-time tracking, we did this by synthesizing a specific DNA aptamer that binds DEHP and Cy5 probes. First, to obtain the DNA aptamer that specifically binds to DEHP, we performed in vitro selection by using Capture-SELEX (systemic evolution of ligands by exponential enrichment) technology, which involves an iterative process of binding, partitioning and amplification (Fig. 1C). After 10 rounds of enrichment, the PCR products were sequenced, resulting in the selection of 25 DNA aptamers as candidates (Figs. 1C and EV2A,B; Appendix Table S3). Meanwhile, the candidates were classified into seven families through gene homology evolution analysis (Fig. EV2C). To determine the aptamer with the highest affinity, we calculated the Gibbs free energy and performed graphene oxide fluorescence assays on these candidates. Ultimately, we selected the DEHP aptamer-Seq.10 based on its low Gibbs free energy ($\triangle$G = −9.9) and dissociation constant ($K_d$ = 0.07 μM, Fig. EV2D; Appendix Table S3). In addition, the results of specificity aligned with the affinity test, further confirming that DEHP aptamer-Seq.10 is the most promising candidate (Fig. EV2E). Based on the simulated secondary structure, aptamer-Seq.10 exhibited stable binding to DEHP through the formation of hydrogen bonds.

To achieve visualization, we conjugated the Cy5 probe to the aptamer through the formation of an amide bond between the carboxylic acid group of the aptamer and the amino group of Cy5. Fluorescence spectral analysis revealed that the aptamer-Seq.10-Cy5 complex displayed absorption peaks at both 650 nm and 670 nm, which overlapped with the absorption spectra of both the aptamer and Cy5. This indicates the successful synthesis of the Cy5-labeled DNA aptamer specific to DEHP (Fig. EV3A). Utilizing the strong affinity and specificity of aptamer-Seq.10, we proceeded to create Cy5-labeled single-stranded DNA aptamer-based DEHP (*csa*-DEHP) by combining the Cy5-labeled DNA aptamer with DEHP. In the $^1$H NMR spectra of DEHP, the peaks at 7.25 ppm and 7.22 ppm were signals of hydrogen corresponding to benzene, while the peaks at 4.2 ppm and 1.6 ppm were signals of methylene groups on the carbon chain. In the aptamer spectrum, the peaks at 3.4 ppm were characterized as signals of methylene groups, while the signals of amino groups were observed at 1.9 ppm. For *csa*-DEHP, both the signals of DEHP's hydrogen and the aptamer's amino groups were similarly detected. However, the signals of the amino groups were chemically shifted to 2.3 ppm, which suggests that DEHP is bound to its aptamer through hydrogen bonding (Figs. 1C and EV3B–D).

CCK-8 analysis was used to evaluate the potential toxicity of this system. As shown in Fig. EV3E, *csa*-DEHP did not have an impact on the viability of mouse PHs. Moreover, *csa*-DEHP remained

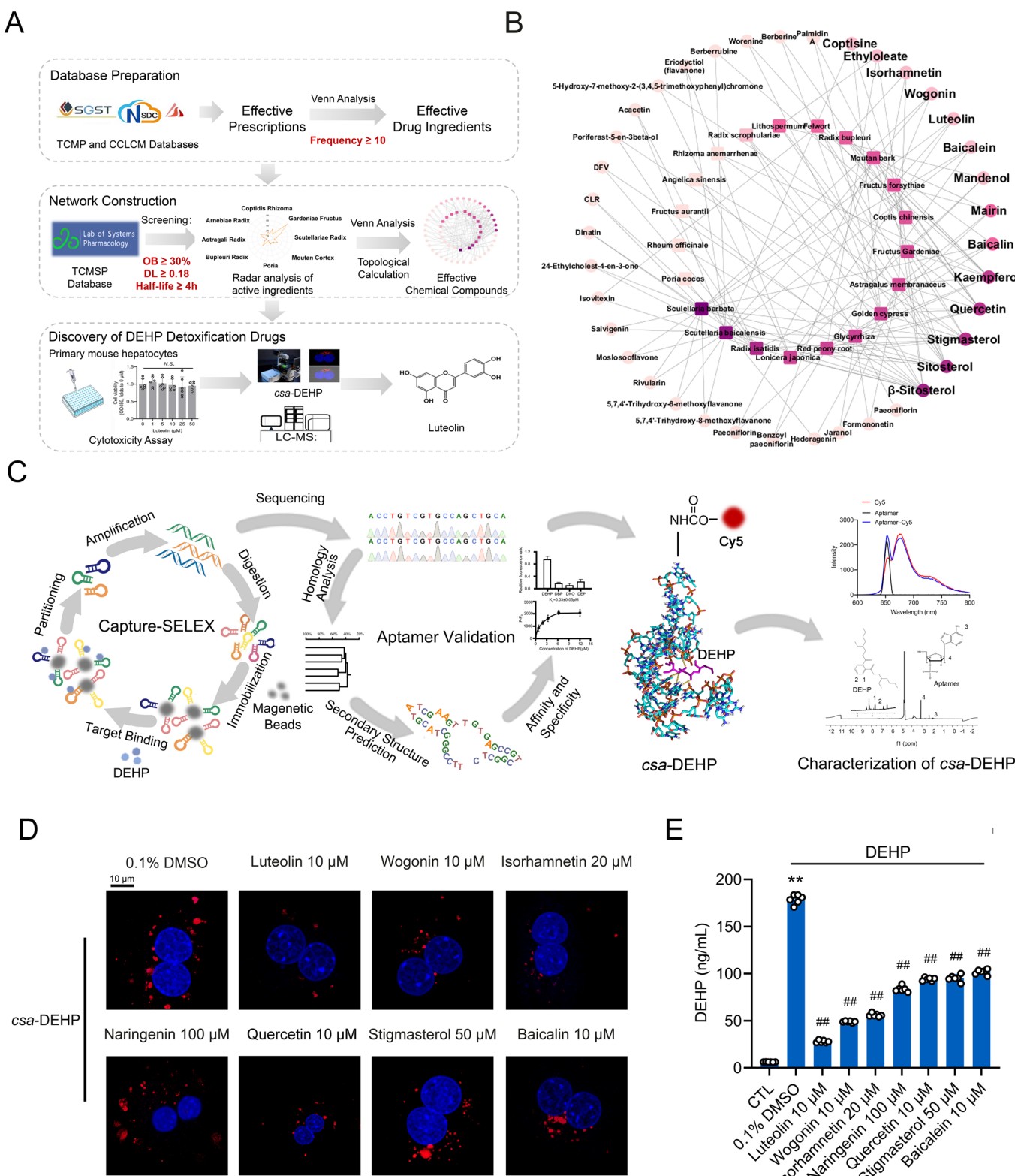

stable in the cell plasma for 48 h, with particular enrichment in the mitochondria and lysosomes (Fig. EV3F–H). To illustrate the cellular uptake mechanisms of csa-DEHP, mouse PHs were treated with csa-DEHP for 12 h in the presence of various inhibitors targeting endocytosis and macropinocytosis. Cellular uptake of csa-

DEHP was significantly reduced by the addition of all inhibitors, including chlorpromazine (a clathrin-mediated endocytosis inhibitor), filipin (a caveolae-mediated endocytosis inhibitor) and EIPA (a macropinocytosis inhibitor) (Fig. EV3I,J). These findings suggest that both endocytosis and macropinocytosis are the dominant

**Figure 1. Network pharmacology analysis for the novel drugs possessing the abilities for the hepatic removal of excess DEHP.**

(A) Schematic diagram illustrating the screening process for the novel drugs that decrease DEHP accumulation in the mouse liver. (B) Topological network analysis of drug ingredients and effective chemical compounds. (C) A Schematic diagram illustrating the detailed synthesis processes for *csa*-DEHP. (D) CLMS analysis of mouse PHs exposed to 150 nM *csa*-DEHP for 12 h, followed by incubation with the indicated drugs for additional 24 h in the absence of *csa*-DEHP to assess the efficacy of candidate drugs in reducing DEHP accumulation. (E) HPLC-MS analysis for the DEHP content retained in mouse PHs exposed to 10 μM DEHP and then similar treated with indicated drugs as described in (D). **$P < 0.01$ *vs.* CTL group, ##$P < 0.01$ *vs.* DEHP + 0.1% DMSO group. $n = 6$. All the data were represented as the mean ± SD. The paired Student's *t*-test was employed to compare between two groups. One-way ANOVA with a Fisher's LSD post hoc test was utilized to compare among multiple groups. Exact *P* values are listed in Appendix Table S11. Source data are available online for this figure.

pathways for the cellular uptake of *csa*-DEHP. In addition, the uptake of *csa*-DEHP was significantly reduced when cells were treated at 4 °C, indicating that temperature may play a role in the uptake process. Since low temperature can alter the physical structure of the lipid bilayer, the inhibition of *csa*-DEHP uptake at 4 °C suggests that *csa*-DEHP still retains its characteristics as a liposoluble substance (Fig. EV3K,L). Strikingly, we observed prolonged cellular retention and strong fluorescent signals of *csa*-DEHP in mouse PHs in comparison to both free Cy5- and aptamer-Cy5-treated groups, indicating that DEHP is a critical factor for the efficacy of the *csa*-DEHP system (Fig. EV3M,N).

In addition, we assessed a long-term stability of the *csa*-DEHP system by using *csa*-DEHP stored at −80 °C for over 1 year. As shown in Appendix Fig. S2A, although fluorescence intensity exhibited a slight decline over time, the overall stability of the system was maintained, as demonstrated by the consistent fluorescence signals retained for more than a year despite the cold storage conditions. The parallel test with ¹H NMR spectra provided additional validation (Appendix Fig. S2B). No significant changes in the spectral characteristics, confirming that the chemical structure of DEHP was preserved without major degradation over the extended period. This result indicates that the stability of the *csa*-DEHP system at −80 °C is suitable for maintaining the quality of its measurements and supports its application in long-term research projects and storage scenarios without compromising accuracy.

To confirm the effectiveness of 7 monomers in promoting the detoxification of excess DEHP in mouse PHs, we exposed mouse PHs to *csa*-DEHP for 12 h and then incubated them with the indicated drugs for 24 h in the absence of *csa*-DEHP. A subsequent confocal laser scanning microscopy (CLMS) analysis revealed a decrease in the fluorescent signals of *csa*-DEHP in response to all of the monomers. Among these, luteolin exhibited the strongest ability to decrease the hepatic retention of DEHP (Fig. 1D; Appendix Fig. S3A). This finding was subsequently validated through HPLC-MS analysis (Fig. 1E).

## Luteolin detoxifies DEHP by facilitating its excretion from the mouse livers

Importantly, luteolin reduced the DEHP accumulation from mouse PHs in a manner that was dependent on both the dosage and duration of treatment (Fig. 2A; Appendix Fig. S3B,C). In addition, to exclude bias, we compared the effects of luteolin on the fluorescent release of Cy5, aptamer-Cy5 and *csa*-DEHP in mouse PHs. As shown in Fig. 2B and Appendix Fig. S3D, luteolin only induced a dramatic fluorescent reduction of *csa*-DEHP without altering the fluorescent signals of either Cy5 or aptamer-Cy5. To

confirm the beneficial effects of luteolin, mice were subjected to a 14-day treatment with DEHP, followed by an additional 14-day luteolin treatment in the presence of DEHP. As shown in Fig. EV4A,B, solitary exposure to DEHP, as well as its synergistic interaction with luteolin, did not yield any deviation from standard mouse body weight or blood glucose levels, maintaining them within normal reference ranges. The meticulous evaluation of blood pressure, through the analysis of serum angiotensin concentrations, further substantiated this stability, with the biomarker consistently present in all experimental cohorts (Fig. EV4C). Therefore, these findings implied that neither compound exerted a substantial impact on the metabolic profile of the test subjects. HPLC-MS analysis revealed that luteolin reduced the levels of DEHP and MEHP (a metabolite of DEHP in the liver) in the mouse liver, serum, urine, as well as white adipose tissue (Figs. 2C,D and EV4D,E). Furthermore, luteolin also alleviated DEHP-induced liver injury, as indicated by the decreased levels of aspartate aminotransferase (AST) and glutamate pyruvic transaminase (ALT) in the serum of mice treated with luteolin (Fig. 2E). Consistently, DEHP-induced macrophage infiltration was decreased in response to luteolin treatment (Fig. 2F,G). In addition, reducing liver accumulation could lead to chronic over-exposure to DEHP/MEHP for the kidneys (Zhang et al, 2023), where they induce epithelial-to-mesenchymal transition (EMT) and fibrosis (Wu et al, 2018), and this could imply damage in terms of filtration rate or albuminuric loss. Hence, to evaluate the potential collateral organ damage of DEHP, we performed histological analyses to investigate the renal injury induced by DEHP. H&E and Masson's trichrome staining analyses revealed that administration of DEHP led to significant renal impairment as evidenced by exacerbated tubular dilation, cast formation, loss of brush borders, and an increased presence of collagen fibers within the kidneys of DEHP-exposed mice (Fig. EV4F). Notably, the renal expression levels of EMT biomarkers, including N-cadherin, Vimentin and α-SMA, exhibited substantial upregulation in response to DEHP exposure (Fig. EV4G). Analogous to its proven hepatoprotective effects, luteolin exhibited a renoprotective effect by mitigating DEHP-induced renal impairment and concomitantly decreasing the protein expression of EMT biomarkers within the kidney of DEHP-treated mice. As a result, our attention and focus have now been directed toward luteolin as our chosen detoxification drug.

## Hepatic Uroc1 is responsive to luteolin treatment at the translational level

Although the beneficial effects of luteolin have been successfully established, the molecular target of luteolin remains unknown. To address this question, we performed label-free proteomics analyses

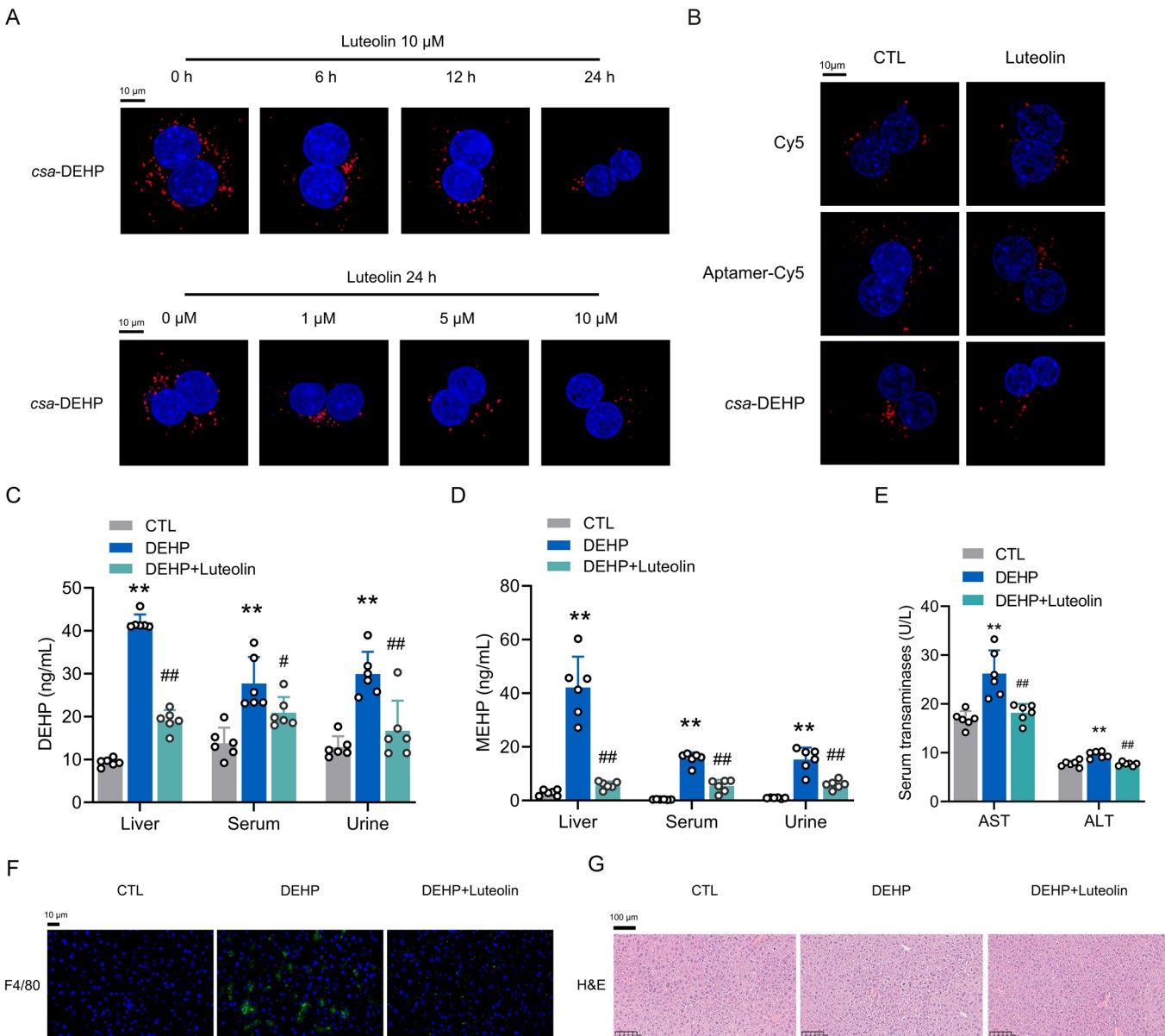

**Figure 2. Luteolin is a potential drug to decrease DEHP accumulation in the mouse liver.**

(A) Representative CLMS images of mouse PHs exposed to 150 nM csa-DEHP for 12 h, followed by with luteolin according to time points and concentrations in the absence of csa-DEHP. (B) Representative CLMS images of mouse PHs treated with 150 nM Cy5, Aptamer-Cy5 and csa-DEHP for 12 h, followed by the luteolin incubation for 24 h in the absence of aforementioned reagents. csa-DEHP, red; Nuclei, blue. Scale bar: 10 µm. (C) HPLC-MS analysis of DEHP levels in the liver, serum, and urine of mice subjected to a 14-day treatment with DEHP (10 mg/kg body weight/day), followed by an additional 14-day treatment with luteolin (10 mg/kg body weight/day) in the presence of DEHP. **$P < 0.01$ vs. CTL group. #$P < 0.05$ and ##$P < 0.01$ vs. DEHP group. $n = 6$. (D) HPLC-MS analysis of MEHP levels in the liver, serum, and urine of mice treated as described in (C). **$P < 0.01$ vs. CTL group. ##$P < 0.01$ vs. DEHP group. $n = 6$. (E) Serological analysis of AST and ALT levels in the serum of mice treated as described in (C). **$P < 0.01$ vs. CTL group. ##$P < 0.01$ vs. DEHP group. $n = 6$. (F) F4/80 staining in liver sections of different mice treated as described in (C). (G) H&E analysis in liver sections of different mice treated as described in (C). All the data were represented as the mean ± SD. The paired Student's $t$-test was employed to compare between two groups. One-way ANOVA with a Fisher's LSD post hoc test was utilized to compare among multiple groups. Exact $P$ values are listed in Appendix Table S11. Source data are available online for this figure.

using samples of mouse PHs. These hepatocytes were treated with DEHP for 12 h, followed by treatment with either luteolin or 0.1% DMSO for another 12 h in the absence of DEHP (Fig. 3A). The differentially expressed proteins were screened out following the criteria that $P < 0.05$ and fold change ≥1.5. As shown in Fig. 3B, a

cluster of 21 proteins was changed in response to luteolin treatment, with 9 upregulated and 12 downregulated. Gene ontology (GO) analysis is closed alterations in biological processes like cellular component synthesis and assembly (Fig. EV5A and Dataset EV3), which is consistent with previous findings that

highlighted luteolin's role in maintaining the cellular homeostasis of hepatocytes (Li et al, 2023). Subsequently, we proceeded to analyze the molecular docking capabilities of luteolin toward these potential targets, ultimately identifying Uroc1 as exhibiting the highest potential for binding affinity, evident from its lowest S value ($S = -7.12$, Fig. 3C; Appendix Table S4). Furthermore, DEHP exhibited modest effects on Uroc1 expression at both the transcriptional and translational levels. Luteolin, on the other hand, had no impact on mRNA expressions, yet led to a decrease in Uroc1 protein levels of Uroc1 in both mouse PHs and livers, by 36.89% and 39.90%, respectively (Figs. 3D–H and EV5B–D). Importantly, a CHX chase experiment revealed that luteolin expedited the protein degradation of Uroc1, resulting in a shortened half-life of 31.7 h (Fig. 3I).

## Hepatic Uroc1 serves as the molecular target of luteolin

To further elucidate the mechanism by which luteolin reduces Uroc1 protein levels, we inhibited the protein degradation pathways using either MG132 (a protein ubiquitination inhibitor) or Baf A1 (an autophagosome-lysosome mediated protein degradation inhibitor). The results suggested that the luteolin-induced protein degradation of Uroc1 was partially inhibited by treatment with MG132, indicating the involvement of protein ubiquitination in this degradation process (Fig. 4A). Moreover, further molecular docking analysis predicted that luteolin bound to Uroc1 at the Ala270 and Val272 sites (Fig. 4B). We then engineered mutant forms of mouse Uroc1 containing either Ala270 and Val272 to Gln mutations or both. In addition, these alternations were predicted to have no impact on the secondary structure of the protein (Fig. EV5E), which was verified by using the SOPMA algorithm. To confirm the essential roles of the Ala270 and Val272 sites, we transfected these mutants into mouse PHs and subjected them to similar treatment as previously mentioned. As shown in Fig. 4C, the luteolin-induced reduction in Uroc1 protein expression levels was partially reversed when the Ala270 or Val272 sites were mutated and completely reversed when both sites were mutated. Accordingly, luteolin treatment led to an increase in the ubiquitination of Uroc1, which was partially reversed by mutations at either the Ala270 or Val272 site and completely abolished by the double mutant.

Functionally, the overexpression of Uroc1 significantly reversed the luteolin-induced release of DEHP in mouse PHs. In contrast, this recapitulation was completely abrogated when Uroc1 was double mutated at both the Ala270 and Val272 sites (Figs. 4D and EV5F). Consistent with these results, knockdown of Uroc1 decreased DEHP accumulation in mouse PHs, as evidenced by the reduced fluorescent signals of csa-DEHP (Figs. 4E and EV5G). An in vivo study revealed that compared with DEHP-fed mice treated with luteolin, the levels of DEHP and MEHP in mouse livers, serum, and urine were significantly restored in mice with adeno-associated virus serotype 8 (AAV8) -mediated Uroc1 overexpression (Figs. 4F,G and EV5H). Meanwhile, serum levels of AST and ALT were similarly increased in these mice (Fig. 4H). Accordingly, similar restoration of undesirable macrophage infiltration was observed in the liver of mice subjected to AAV8-Uroc1 injection (Fig. 4I). These results further emphasized our conclusion that Uroc1 is the specific target of luteolin, while the Ala270 or Val272 sites function as key amino acid sites involved in facilitating

the protein degradation, subsequently contributing to the luteolin-induced degradation of Uroc1.

## *Trans*-UCA serves as the functional metabolite mimicking luteolin

Given the existence of multiple molecular targets for luteolin and considering the time-consuming and costly process of de novo chemical drug development targeting Uroc1, our focus was directed toward the endogenous metabolites that are correlated to Uroc1. Since Uroc1 is an enzyme that catalyzes the transformation of *trans*-UCA into 4-imidazolone-5-propanoate, a decrease in Uroc1 protein levels results in an increase in its substrate *trans*-UCA, as well as the upstream substrate (Fig. 5A). These potential endogenous metabolites are believed to trigger the hepatic release of DEHP. To test this hypothesis, we performed a CCK-8 assay to assess the safe doses of these two metabolites for mouse PHs (Appendix Fig. S4A–D). We found that both L-histidine (L-his) and *trans*-UCA exhibited no toxicity on cell viability, even in the presence of DEHP. Further CLMS analysis based on *csa*-DEHP revealed that *trans*-UCA, but not L-his, significantly decreased the fluorescent signals of *csa*-DEHP (Fig. 5B; Appendix Fig. S4E). This finding was further validated by LC–MS analysis. Although L-his decreased the DEHP content by 50.22%, *trans*-UCA exhibited effects similar to those of luteolin by decreasing the DEHP levels by 84.48% (Fig. 5C). Accordingly, *trans*-UCA reduced DEHP accumulation in mouse PHs in a manner that is dependent on both time and dose (Fig. 5D; Appendix Fig. S4F,G). Surprisingly, *trans*-UCA was comparable to luteolin in vivo and reduced the levels of DEHP and MEHP in mouse livers, serum, and urine (Fig. 5E,F), while also decreasing the DEHP-induced increase in serum AST and ALT, as well as the macrophage infiltration (Fig. 5G,H).

## ERK1/2-orchestrated lysosome pathway is involved in mediating the luteolin/*trans*-UCA-facilitated hepatic DEHP release

To investigate the signaling pathways that mediate *trans*-UCA signals, we performed label-free proteomics analyses using samples of mouse hepatocytes that were subjected to DEHP for 12 h and then either treated with *trans*-UCA or PBS for an additional 12 h without DEHP (Fig. 6A). The differentially expressed proteins were screened out following the criteria that $P < 0.05$ and fold change $\geq 2$. A total of 76 proteins were altered by *trans*-UCA treatment, with 27 upregulated and 49 downregulated (Fig. 6B). GO analysis indicated that these proteins were clustered in the MAPK pathways, particularly in the pathways associated with ERK1/2 (Fig. 6C and Dataset EV4). We found that DEHP significantly elevated the phosphorylation levels of ERK1/2 proteins. In contrast, both *trans*-UCA and luteolin exhibited a remarkable ability to reverse the ERK1/2 phosphorylation induced by DEHP (Fig. 6D).

It has been documented that lysosome-associated pathways are key processes for the excretion of toxins and xenobiotics (Persy et al, 2006). Since this pathway is tightly regulated by ERK1/2 proteins (Chen et al, 2017), we hypothesized that the ERK1/2-orchestrated lysosome pathway is a key route for eliminating excessive DEHP. By utilizing Baf A1, we found that inhibition of

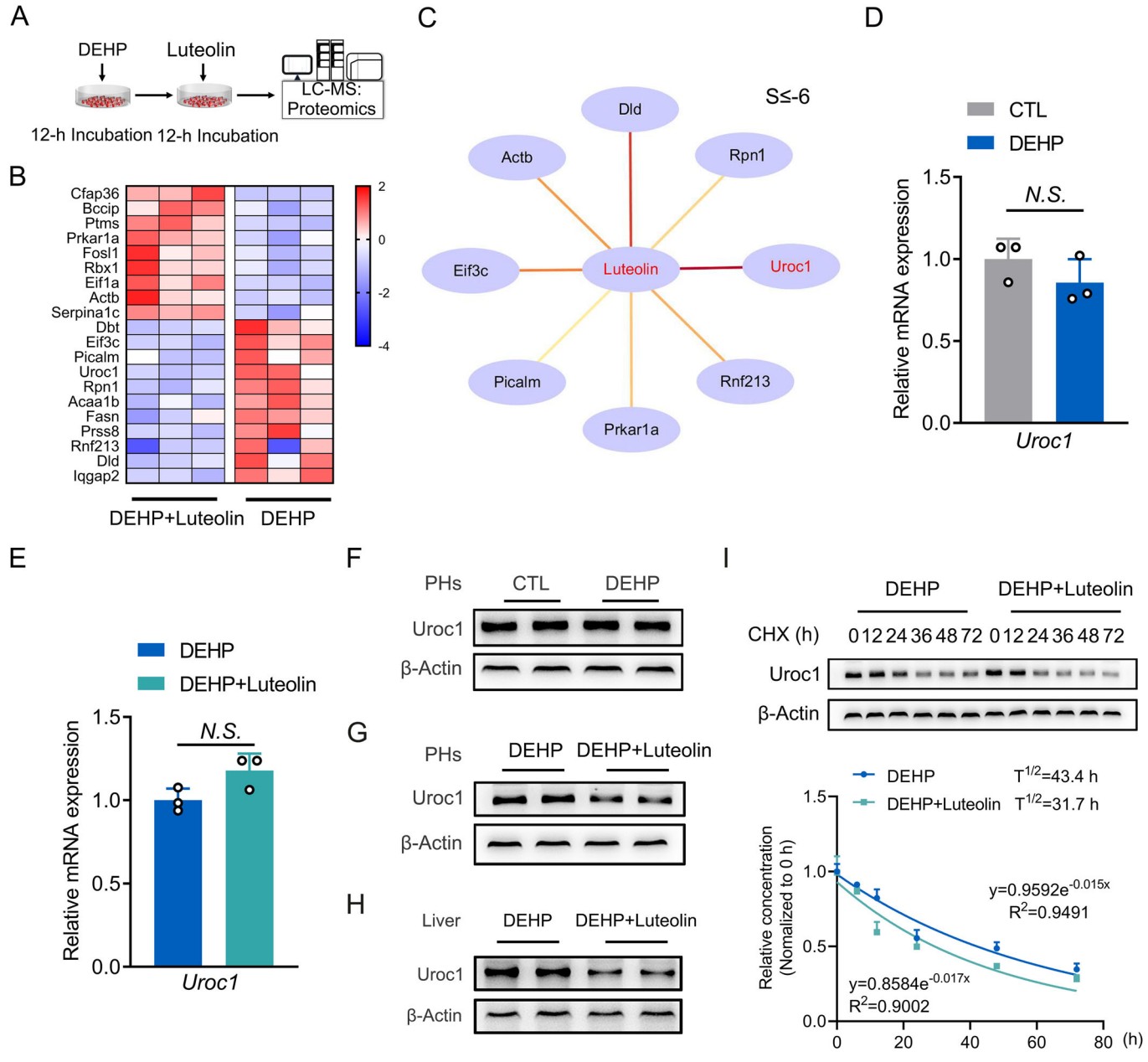

**Figure 3. Hepatic Uroc1 is responsive to luteolin treatment at the translational level.**

(A) Schematic diagram illustrating the label-free proteomics analyses used for the molecular target scanning of luteolin. (B) Heatmap analysis of differentially expressed proteins. (C) MOE analysis of binding potentials of luteolin on differentially expressed proteins. The proteins with the S value ≤ −6 was presented. (D) RT-qPCR analysis of Uroc1 expression in mouse PHs exposed to 10 μM DEHP for 12 h. *N.S.*, no significance. $n = 3$. (E) RT-qPCR analysis of Uroc1 expression in mouse PHs that were similarly treated as described in Fig. 2B. *N.S.*, no significance. $n = 3$. (F) Western blot analyses of Uroc1 expression in mouse PHs that were similarly treated as described in (D). (G) Western blot analyses of Uroc1 expression in mouse PHs that were similarly treated as described in (E). (H) Western blot analyses of Uroc1 expression in the liver of mice treated as described in Fig. 2C. (I) CHX chase analysis for the half-life of Uroc1 protein. $n = 3$. '$n$' represents biological replicates. All the data were represented as the mean ± SD. The paired Student's *t*-test was employed to compare between two groups. Exact $P$ values are listed in Appendix Table S11. Source data are available online for this figure.

lysosome activity greatly negated the beneficial effects of both luteolin and *trans*-UCA on the hepatic clearance of DEHP (Fig. 6E,F; Appendix Fig. S4H). Taking into account the previous subcellular distribution of *csa*-DEHP, which was concentrated in the lysosome, this organelle is an essential mechanism per se for DEHP clearance.

# Discussion

DEHP was listed as an environmental endocrine disrupting chemical in the Endocrine Society's first scientific statement in 2009 and gained further prominence in its second scientific statement in 2015 due to its pathological effects on metabolic

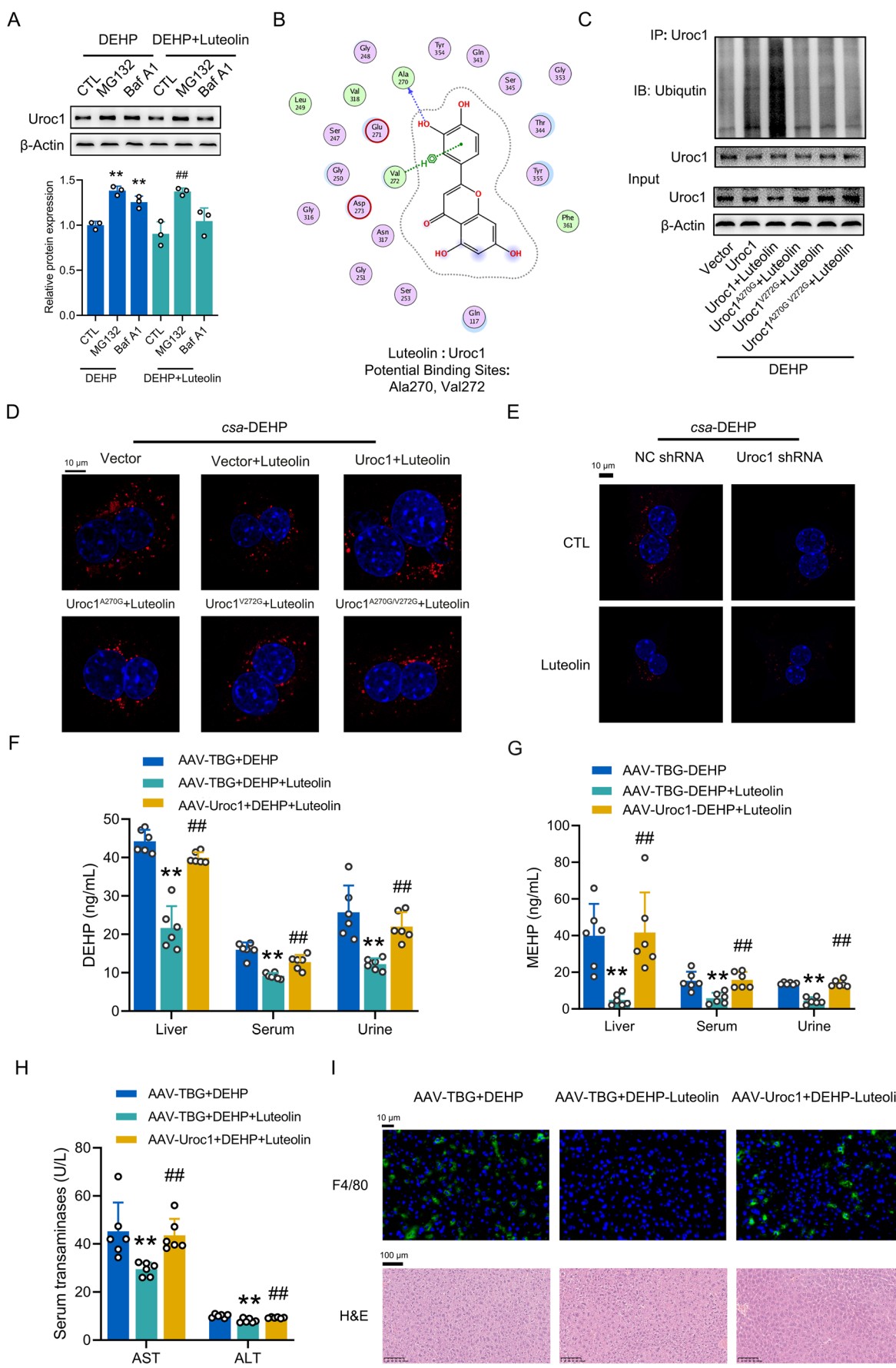

**Figure 4. Hepatic Uroc1 serves as the molecular target of luteolin.**

(A) Western blot analysis of Uroc1 expression in mouse PHs treated with 10 μM DEHP for 12 h, followed by a single stimulation with 10 μM luteolin for 6 h in the absence of DEHP. Subsequently, these cells were further treated with a combination of 10 μM luteolin and 10 μM MG132 or 20 nM Baf A1 for 6 h. **$P < 0.01$ vs. DEHP group, ##$P < 0.01$ vs. DEHP+Luteolin group. $n = 3$. (B) Molecular docking analysis of mouse Uroc1 and luteolin. (C) Western blot analyses of ubiquitination of Uroc1 protein expression and in mouse PHs. (D) Representative CLMS images of csa-DEHP in mouse PHs transfected with indicated plasmids for 48 h, followed by a 12-h incubation with csa-DEHP. These cells were subsequently treated by luteolin for another 24 h in the absence of csa-DEHP. (E) Representative CLMS images of csa-DEHP in mouse PHs transfected with either Uroc1 shRNA or NC shRNA for 48 h, which were further treated as described in (D). csa-DEHP, red. Nuclei, blue. Scale bar: 10 μm. (F) HPLC-MS analysis of DEHP levels in the liver, serum, and urine of mice with AAV8-mediated Uroc1 overexpression or their corresponding controls, treated similarly as described in Fig. 2C. (G) HPLC-MS analysis of MEHP levels in the liver, serum, and urine of mice treated as described in (F). (H) Serological analysis of AST and ALT levels in the serum of mice treated as described in (F). (I) F4/80 staining (top) and H&E analysis (down) in liver sections of different mice treated as described in (F). **$P < 0.01$ vs. AAV-TBG + DEHP group. ##$P < 0.01$ vs. AAV-TBG + DEHP+Luteolin group. $n = 6$. All the data were represented as the mean ± SD. One-way ANOVA with a Fisher's LSD post hoc test was utilized to compare among multiple groups. Exact $P$ values are listed in Appendix Table S11. Source data are available online for this figure.

diseases, including nonalcoholic fatty liver diseases and atherosclerosis (Diamanti-Kandarakis et al, 2009; Gore et al, 2015). Classic drugs, such as icariin and lycopene, which exhibit antioxidative or anti-inflammatory properties, have demonstrated efficacy in alleviating the toxic repercussions of DEHP (Sun et al, 2022; Zhao et al, 2022). Nevertheless, considering the inevitable exposure to DEHP in the external environment, its detrimental effects on human health continue to persist. The aforementioned medications merely offer transient alleviation without addressing the underlying issue of DEHP. Moreover, the direct molecular targets and the true molecular status of DEHP remain elusive, posing significant challenges and puzzlement in addressing the health concerns associated with it. In light of this, our study adopted a different approach by focusing on DEHP as a target, aiming to identify medications that can enhance its release from the body and reduce its duration of stay, thereby ultimately mitigating its detrimental effects on human health. Leveraging our self-developed visual DEHP tracing system and employing network pharmacology analysis approaches, our study revealed that luteolin exhibits significant efficacy in reducing DEHP accumulation in both mouse PHs and livers. This beneficial effect of luteolin is achieved by enhanced protein degradation of hepatic Uroc1 and induction of trans-UCA-driven lysosomal exocytosis, facilitating the excretion of the toxic compound. Our findings highlight Uroc1 as a novel therapeutic target for the detoxification of DEHP and related contaminants in the liver. Furthermore, luteolin and endogenous trans-UCA show promise as functional drugs and metabolites, respectively.

DEHP, with its persistence in the food chain, is commonly found in contaminated oil crops and animal fats, as well as in plastic packaging where it leaches into consumables during storage (Chen et al, 2012; Gao et al, 2023). Oral intake, especially from contaminated food, accounts for over 90% of DEHP absorption in humans, which underscores the significance of the liver as the primary site of metabolism for this compound (Li et al, 2022). Our study's focus on oral ingestion and the liver as the target, addressing the potential of luteolin and its derivative trans-UCA in reducing DEHP and MEHP accumulation within the liver. However, it's crucial to acknowledge that DEHP's entry into the body takes various forms, such as dermal absorption, intravenous administration, and inhalation, each with its own specific organ distribution patterns (Li et al, 2022). The organs, such as pancreas and spleen, particularly in cases of dermal penetration, exhibit higher concentrations. Therefore, future research investigating the detoxifying effects of luteolin in these alternative sites would

broaden its applicability in against DEHP exposure through different routes, making it a more versatile strategy for combating environmental pollutants. These studies will contribute to a more comprehensive and generalizable understanding of luteolin's protective role against DEHP toxicity.

The elusive nature of DEHP presents challenges in its detection, leading to a lack of efficient high-throughput screening techniques and a limited selection of drugs capable of specifically targeting and facilitating the release of DEHP. To address this problem, the autoradiography technique was previously employed to track the distribution of DEHP in the rat testis using ³H-labeled DEHP (Ono et al, 2004). However, this method called for an extended period of development for the tested sections. In addition, the utilization of the autoradiography technique might entail the use of specialized instruments or devices, along with the implementation of additional safety precautions to safeguard experimenters. Moreover, due to the absence of a carboxyl group (-COOH), DEHP differs from its metabolite Mono-2-ethylhexyl phthalate (MEHP), which readily forms cross-links with the amino group (-NH₂) present on the fluorescent probes (Huo et al, 2019). Therefore, the creation of an innovative visual detection and tracking system for DEHP is of paramount importance and urgency. To achieve the visualized DEHP tracing system, we designated a Cy5-labeled aptamer targeting DEHP. In recent years, significant progress has been made in generating DNA and RNA aptamers against a diverse array of targets, including small molecules, chemical compounds, proteins, and various other biomolecules (Zhou and Rossi, 2017). Aptamers possess a remarkable advantage in terms of their simple and cost-efficient synthesis, enabling convenient chemical modifications (Liu et al, 2023). In addition, aptamers exhibit the remarkable ability to undergo conformational changes upon interacting with their specific targets (Tsukakoshi et al, 2010). Despite their ability to display a high degree of affinity and specificity toward their targets, aptamers showcase a myriad of captivating attributes that confer upon them considerable value as molecular recognition elements. In this study, we broadened the application of aptamers to visualize DEHP, thereby achieving real-time monitoring of DEHP in mouse PHs. In comparison to conventional chemical modifications of molecules, this approach offers a novel strategy for the real-time tracking of small molecular drugs and contaminants. This advancement contributes significantly to the investigation of pharmacodynamics and toxicodynamics, as well as the screening of targeted drugs. Furthermore, the aptamer structure lends itself to easy modification, enabling efficient and controllable formation of multiple drug structures

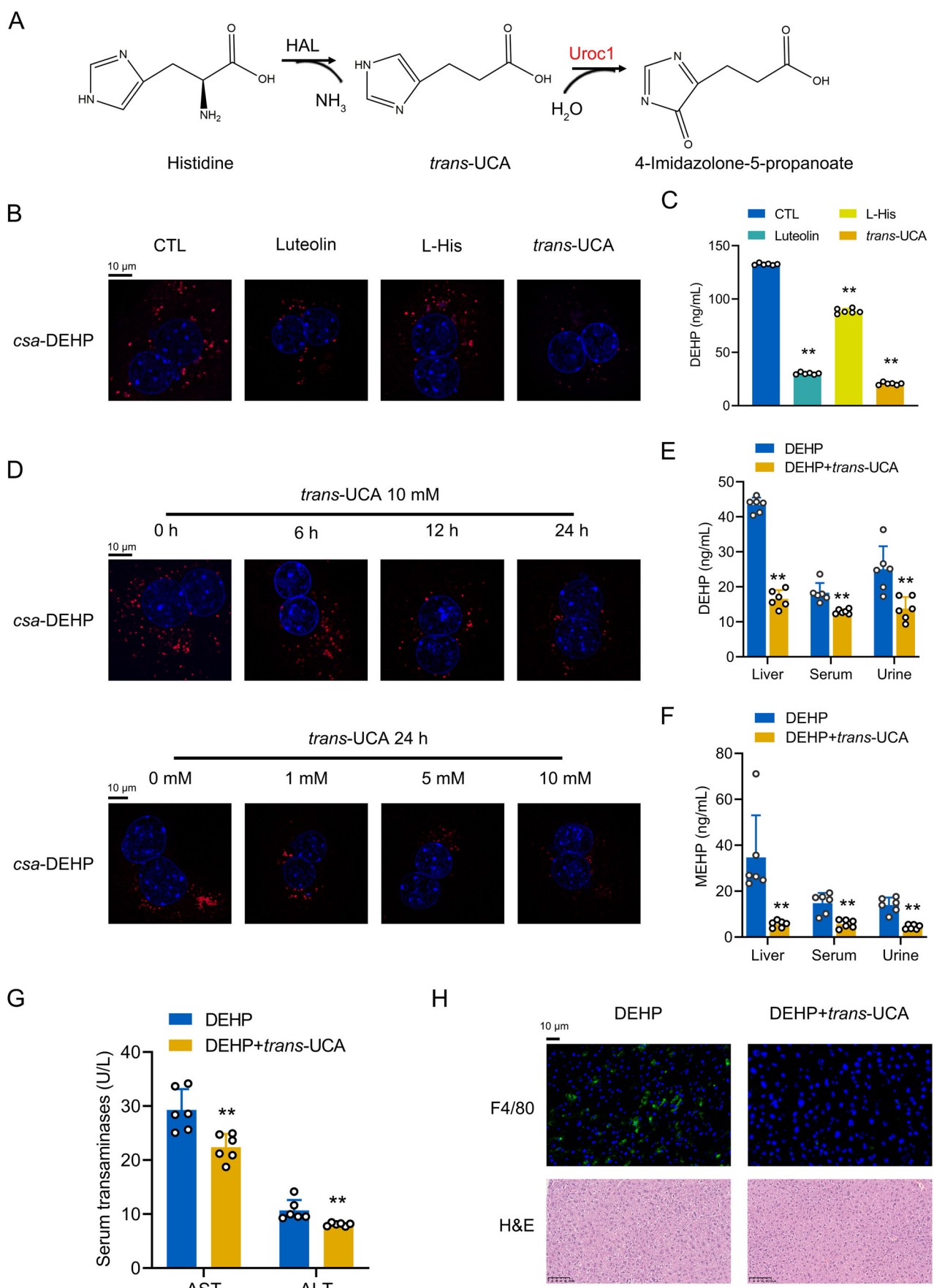

**Figure 5.  *Trans*-UCA serves as the functional metabolite mimicking the luteolin.**

(**A**) Schematic diagram illustrating the role of Uroc1 in histidine metabolism. (**B**) CLMS analysis of mouse PHs treated with 150 nM *csa*-DEHP for 12 h, followed by incubation with 10 μM luteolin, 100 μM L-his, 10 mM *trans*-UCA, respectively, for 24 h in the absence of *csa*-DEHP. (**C**) HPLC-MS analysis for the DEHP content retained in mouse PHs exposed to 10 μM DEHP and subsequently treated with the indicated drugs as described in (**B**). **$P < 0.01$ *vs.* CTL group. $n = 6$. '*n*' represents biological replicates. (**D**) Representative CLMS images of mouse PHs were exposed to 150 nM *csa*-DEHP for 12 h, followed by *trans*-UCA according to time points and concentrations in the absence of *csa*-DEHP. (**E**) HPLC-MS analysis of DEHP levels in the liver, serum, and urine of mice subjected to a 14-day treatment with DEHP (10 mg/kg body weight/day) followed by an additional 14-day *trans*-UCA (10 mg/kg body weight/day) treatment in the presence of DEHP. $n = 6$. '*n*' represents biological replicates. (**F**) HPLC-MS analysis of MEHP levels in the liver, serum, and urine of mice treated as described in (**E**). $n = 6$. '*n*' represents biological replicates. (**G**) Serological analysis of AST and ALT levels in the serum of mice treated as described in (**E**). $n = 6$. '*n*' represents biological replicates. (**H**) F4/80 staining (top) and H&E analysis (down) in liver sections of different mice treated as described in (**E**). **$P < 0.01$ *vs.* DEHP group. $n = 6$. All the data were represented as the mean ± SD. One-way ANOVA with a Fisher's LSD post hoc test was utilized to compare among multiple groups. Exact *P* values are listed in Appendix Table S11. Source data are available online for this figure.

while also enhancing water solubility (Li et al, 2017). Significantly, aptamers demonstrate excellent tissue penetration and possess favorable in vivo safety profiles without eliciting obvious immune responses (Ma et al, 2021). Of note, there are two critical factors to consider in relation to the cellular absorption of DEHP. First, DEHP is a lipophilic compound with a small molecular size, which implies that its entry into cells can potentially transpire through a lipid bilayer interaction mechanism on the cellular membrane, analogous to compatibility (Kohda et al, 2012). Second, as a small molecule compound, DEHP has the ability to interact with various intricate proteins present in plasma, such as alpha-1-acid glycoprotein, thereby elevating its molecular weight (Ingram et al, 2018). The transportation of these DEHP molecules, encapsulated within larger biological entities, into cells may potentially take place through ATP-dependent endocytosis. In our study, *csa*-DEHP entered cells via both endocytic processes and the presence of a lipid bilayer on the cell membrane, thereby partially simulating the pathological state of DEHP within the body. On the other hand, drawing inspiration from the highly promising characteristics of aptamers in targeted cancer therapy (Sun et al, 2014), drugs designed based on the combination of aptamer-targeting contaminants and potential metabolites targeting toxin excretion (such as *trans*-UCA) could be developed for the advanced treatment of diseases resulting from environmental contamination.

Luteolin, a natural flavonoid compound, has gained considerable attention in recent years due to its diverse biological activities and potential health benefits. Clinically, luteolin formulation (NeuroProtek, which is approved by the U.S. Food and Drug Administration) containing 100 mg luteolin was found to inhibit mast cells and reduce brain inflammation in children with autism spectrum disorders ($n = 37$; 4–14 years old), indicating the biosafety of luteolin (Wang et al, 2021b). Functionally, one of the key features of luteolin is its ability to exhibit strong antioxidant and anti-inflammatory properties (Gendrisch et al, 2021a). It scavenges free radicals and reactive oxygen species (ROS) and inhibits the production and elimination of proinflammatory cytokines and enzymes in the body, thereby reducing oxidative stress and alleviating systemic inflammation (Hunt et al, 2006; Wolfle et al, 2012). As a result, luteolin has been implicated in the prevention and treatment of various oxidative stress- and inflammation-related diseases, such as cardiovascular diseases, neurodegenerative disorders, and cancers (Gendrisch et al, 2021a; Gendrisch et al, 2021b). Considering the DEHP-induced generation of ROS and inflammation (Wolfle et al, 2012; Zhao et al, 2021), luteolin potentially exerts its beneficial effects by modulating oxidative stress and inflammation. However, in our study,

luteolin exhibited only a marginal impact on these two pathways when acting as a singular molecule reducing DEHP accumulation from mouse PHs. This finding implies that while the detrimental effects of DEHP and the beneficial effects of luteolin are mediated by ROS and inflammation pathways, these pathways are not directly implicated in the clearance of DEHP from the liver. Besides, we observed a renoprotecitve effects of luteolin in DEHP-treated mice. However, the pivotal question that remains to be elucidated is the interdependence: whether the renal protective effect of luteolin is intrinsically linked to its facilitation of hepatic DEHP detoxification or constitutes an independent protective mechanism. This inquiry is of paramount importance for uncovering the intricate molecular interactions and therapeutic implications of luteolin's targets.

On the other hand, as a TCM monomer, luteolin impacts multiple molecular targets, such as Nrf2, nuclear factor-kappa B (NF-κB), and AP-1, which are crucial in mediating its pharmacological effects on diverse ailments (Aziz et al, 2018; Wruck et al, 2007). It has been observed that these molecular targets serve as mediators for the effects of luteolin rather than engaging in direct interactions with it. The precise molecular targets of luteolin remain a topic of ongoing debate. Herein, taking advantage of proteomics and MOE analyses, we successfully identified Uroc1 as a bona fide molecular target of luteolin. Importantly, luteolin initiates protein ubiquitination and subsequent degradation of Uroc1 by binding to specific sites, namely, Ala270 and Val272. Meanwhile, the abrogation of the beneficial effects of luteolin was observed upon mutations at both sites, indicating their indispensable role in maintaining Uroc1 protein stability and facilitating its physiological functions. Notably, Lys274 and Lys279 were found to be located in close proximity to these two sites, suggesting that luteolin's targeting of both sites could potentially contribute to the induction of structural changes, thus exposing these lysine sites for subsequent protein ubiquitination. Furthermore, luteolin demonstrated a pronounced effect in restraining the accumulation of DEHP and MEHP within adipose tissue. Despite this, considering the distinctive protein expression of Uroc1 within the liver (https://www.proteinatlas.org/ ENSG00000159650-UROC1), this reduction could be predominantly attributed to luteolin's potent systemic protective mechanisms when mitigating the hepatotoxic effects of DEHP.

Considering the multitude of targets associated with luteolin and the challenges in developing novel drugs that specifically target Uroc1, it is particularly intriguing to assess the potential advantages of utilizing *trans*-UCA for reducing DEHP accumulation. The rationale behind this stems from the fact that *trans*-UCA

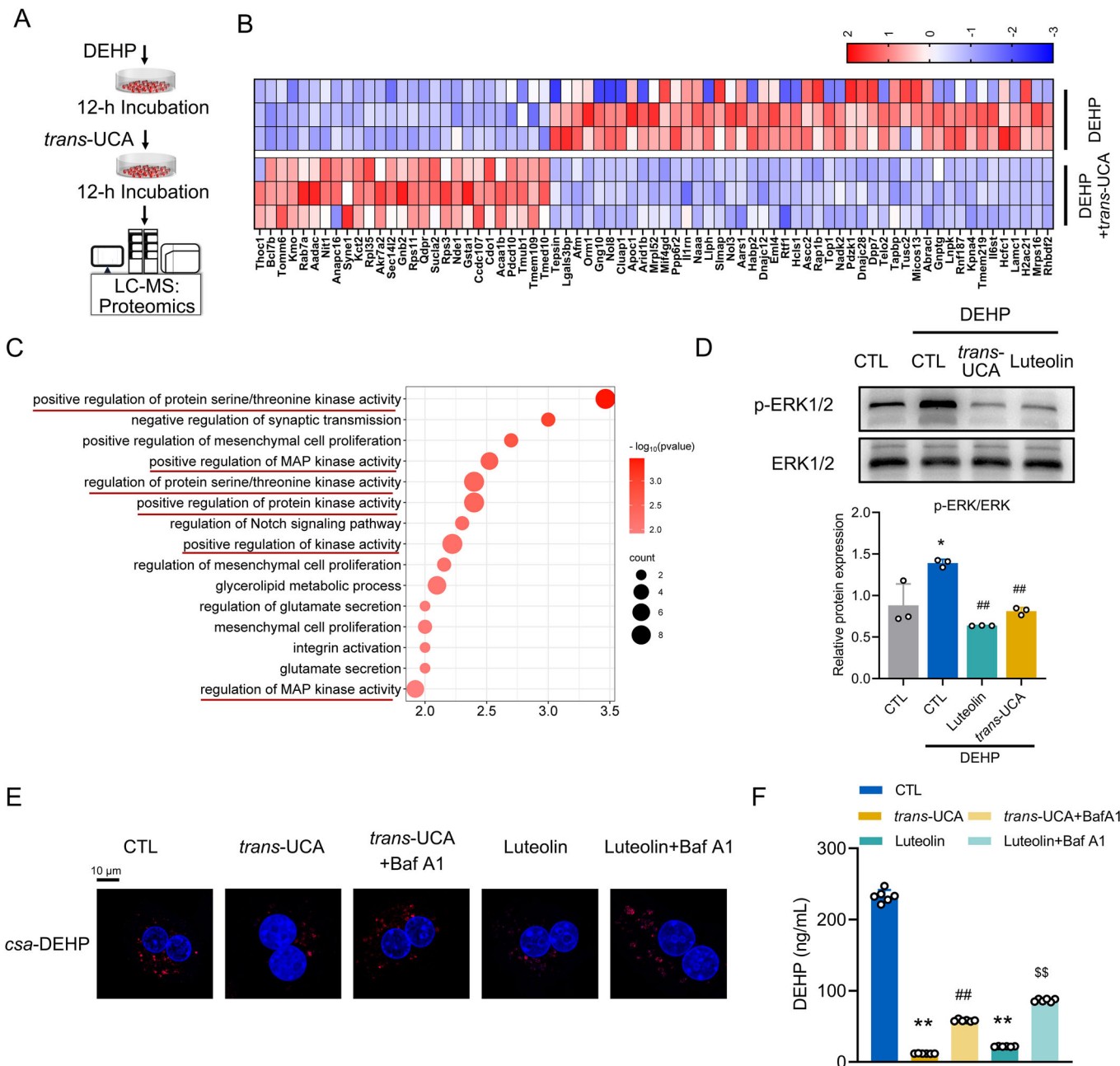

**Figure 6. ERK1/2-orchestrated lysosome pathway is involved in mediating the luteolin/*trans*-UCA-facilitated hepatic DEHP excretion.**

(A) Schematic diagram illustrating the label-free proteomics analysis for the underlying molecular pathways involved in the *trans*-UCA-driven removal of DEHP in mouse PHs. (B) Heatmap analysis of differentially expressed proteins. (C) GO enrichment analysis of differentially expressed proteins (biological process). The GO functional annotations of all differentially expressed proteins were compared with those of the reference species (or all experimentally identified proteins). A Fisher's Exact Test was performed to determine the significance of the differences and identify enriched functional categories (with a P value < 0.05). (D) Western blot analyses of ERK1/2 protein phosphorylation levels in mouse PHs pretreated with either *trans*-UCA or luteolin for 12 h followed by a 15-min DEHP stimulation. **P < 0.05 vs. CTL group, ##P < 0.01 vs. DEHP group. n = 3. (E) Representative CLMS images of *csa*-DEHP in mouse PHs treated with 150 nM *csa*-DEHP for 12 h and then incubated with a combination of Baf A1 and either 10 mM *trans*-UCA or 10 μM luteolin for 24 h in the absence of *csa*-DEHP. (F) HPLC-MS analysis for the DEHP content retained in mouse PHs exposed to 10 μM DEHP, followed by similar treatment as described in (E). **P < 0.01 vs. CTL group, ##P < 0.01 vs. *trans*-UCA group, $$P < 0.01 vs. luteolin group. n = 6. All the data were represented as the mean ± SD. The paired Student's *t*-test was employed to compare between two groups. One-way ANOVA with a Fisher's LSD post hoc test was utilized to compare among multiple groups. Exact P values are listed in Appendix Table S11. Source data are available online for this figure.

functions as a substrate for Uroc1 and is an inherent metabolite demonstrating minimal toxicity at physiologically relevant levels. In our study, we found that *trans*-UCA exerted properties comparable to those of luteolin. It exhibited no toxicity toward the cell viability of mouse PHs while also effectively reducing serum AST and ALT levels, as well as macrophage infiltration in the liver of mice, even at a dosage of 10 mg/kg These findings suggest the safety and advantageous properties of this endogenous metabolite, particularly in relation to hepatocellular function. Furthermore, although luteolin did not show the same Top15 pathways as *trans*-UCA, the MAPK pathways were notably enriched, indicating an involvement in the luteolin-regulated network (Highlighted in Dataset EV3). The discrepancy in the top pathways could be attributed to the differential nature of luteolin and *trans*-UCA. Luteolin, being a small molecule, engages with a diverse range of targets due to its functional groups, while *trans*-UCA as an endogenous metabolite might yield more specific downstream pathways in mitigating DEHP accumulation. In addition, the inverse changes in Acaa1b protein expression could result from distinct responses to the two compounds. Luteolin's possible effect on decreasing Acaa1b might stem from its lipids-lowering properties, which could promote the expression of other lipid β-oxidation enzymes, like FASN (decreased to 57.9% compared to DEHP-treated group), thus leading to a feedback mechanism to decrease Acaa1b protein expression. In contrast, *trans*-UCA's observed increase in Acaa1b could be related to enhanced fatty acid oxidation to fuel histidine metabolism and detoxification of DEHP. Considering that we focused on detoxifying effects of luteolin and *trans*-UCA, this inconsistent change of Acaa1b did not alter the net efficacy of either luteolin and *trans*-UCA, as evidenced by the MAPK pathways were enriched by GO analysis of proteomics results from these two compounds-treated mouse PHs. Despite these inconsistencies, the enriched MAPK pathways in both luteolin and *trans*-UCA treatments suggest that they maintain their overall efficacy in detoxifying processes.

Besides, we noticed that *cis*-UCA has been documented to have adverse effects on the integumentary system, particularly on keratinocytes. *Cis*-UCA functions as an immune modulator generated in the stratum corneum and has a reactive response to ultraviolet (UV) light, prompting its immunosuppressive signals through binding with the $5$-$HT_{2A}$ receptor (Walterscheid et al, 2006). In addition, various cytokines (including TNF-α, PGE2, and IL-10) and ROS signals have been implicated (Kaneko et al, 2008; Shreedhar et al, 1998), further suggesting *cis*-UCA's role as an inducer of inflammation and ROS production. Mechanistically, similar to UV radiation, *cis*-UCA stimulated the ERK signaling pathways necessary for PGE2 synthesis in human keratinocytes (Kaneko et al, 2008; Kaneko et al, 2011). In contrast, our proteomics analysis detected decreased protein expression levels of inflammation-associated proteins, including *IL1rn* and *Orm1*, following *trans*-UCA treatment. Moreover, we demonstrated that *trans*-UCA suppressed the phosphorylation of ERK1/2 caused by DEHP exposure. The disparate outcomes could potentially arise from different types of UCA. It is important to note that UV irradiation is the primary factor responsible for the conversion of *trans*-UCA to *cis*-UCA (Walterscheid et al, 2006). However, as the liver lacks the ability to sense UV light, we believed that *trans*-UCA is the genuine metabolite responsible for decreasing DEHP accumulation. In addition, *cis*-UCA demonstrates significant cytotoxicity at a concentration of 70 μM, whereas *trans*-UCA is substantially safer even at a dosage of 10 mM for mouse PHs, indicating that *trans*-UCA is a safer reagent for systemic home-ostasis (Kaneko et al, 2009; Viiri et al, 2009).

It is important to note that the research approach of "treating symptoms with drugs" contrasts with the principle of "treating diseases with targeted drugs". It neglects the true molecular targets of pollutants present in the body, as well as the multitarget characteristics displayed by individual components of TCM. In this study, we demonstrate that Uroc1 holds immense potential as a therapeutic target for effectively detoxifying DEHP and its associated pollutants in the liver. Of note, luteolin and its related endogenous *trans*-UCA have been identified as potential functional drugs and metabolites, respectively, in this process.

# Methods

**Reagents and tools table**

| Reagent/Resource | Reference or Source | Identifier or Catalog Number |
|---|---|---|
| **Experimental Models** | | |
| Male C57BL/6J | GemPharmatech | N000013 |
| **Recombinant DNA** | | |
| **Antibodies** | | |
| Mouse anti-β-Actin | Proteintech | 66009-1 |
| Rabbit anti-UROC1 | Proteintech | 25032-1-AP |
| Rabbit anti-ubiquitin | Proteintech | 10201-2-AP |
| Rabbit anti-ERK1/2 | Proteintech | 11257-1-AP |
| Rabbit anti-phospho-ERK1/2 | Proteintech | 28733-1-AP |
| Rabbit anti-N-Cadherin | Proteintech | #13116 |
| Rabbit anti-Vimentin | Proteintech | #5471 |
| Rabbit anti-α-SMA | Proteintech | 14395-1-AP |
| Goat anti-mouse HRP | Proteintech | RGAM001 |
| Goat anti-rabbit HRP | Proteintech | RGAR001 |
| **Oligonucleotides and other sequence-based reagents** | | |
| SELEX ssDNA nucleotides | | Appendix Table S5 |
| qPCR primer/probe sets | | Appendix Table S10 |
| **Chemicals, Enzymes and other reagents** | | |
| Lab Rat and Mouse Maintenance Diet | Xietong Bio-engineering | 1010001 |
| Irradiatied Lab Corncob Bedding | Xietong Bio-engineering | 1060016 |
| DMEM, high glucose | Gibco | 11965092 |
| Fetal bovine serum (FBS) | Royacel | RY-F22 |
| Type IV collagenase | Sigma | C4-28 |
| Penicillin-Streptomycin, liquid | Gibco | 15140122 |
| EGTA | Sangon | A600077 |

| Reagent/Resource | Reference or Source | Identifier or Catalog Number |
|---|---|---|
| HEPES | Gibco | 15630080 |
| EDC | Sangon | C600433 |
| Cy5 | APExBIO | A8102 |
| DAPI | Beyotime | C1002 |
| Di(2-ethylhexyl) phthalate (DEHP) | Aladdin | D109649 |
| Mono(2-ethylhexyl) phthalate (MEHP) | Aladdin | M114127 |
| DEHP-D4 | CATO | N/A |
| MEHP-D4 | CATO | N/A |
| Methyl alcohol | Sigma | 34860 |
| Acetonitrile | Sigma | 34851 |
| Ammonium acetate | Merck | 543834 |
| Corn oil | Beyotime | ST1177 |
| Luteolin | J&K | 128789 |
| *trans*-UCA | Sangon | A420053 |
| Cycloheximide (CHX) | Sigma | 239764 |
| MG132 | Sigma | M8699 |
| Bafilomycin A1 (Baf A1) | MCE | HY-100558 |
| Glycine | Sangon | A610236 |
| Tris (hydroxymethyl) aminomethane | Sangon | A610195 |
| Sodium dodecyl sulfate (SDS) | Sangon | A600485 |
| Triton X-100 | Solarbio | T8200 |
| Agarose | Biosharp | BS081 |
| Aspartate aminotransferase Assay Kit | Jiancheng Institute of Biotechnology | C010-2-1 |
| Alanine aminotransferase Assay Kit | Jiancheng Institute of Biotechnology | C009-2-1 |
| Glucose kit (glucose oxidase method) | Jiancheng Institute of Biotechnology | A154-1-1 |
| Angiotensin (ANG) assay kit | COIBO BIO | CB12202-Mu |
| **Software** | | |
| Prism | https://www.graphpad.com/ | N/A |
| FIJI | https://imagej.net/software/fiji/ | N/A |
| MOE | https://www.chemcomp.com/en/Products.htm | N/A |
| Cytoscape | https://cytoscape.org/ | N/A |
| Snapgene | https://www.snapgene.com/ | N/A |
| DNAMAN | https://www.lynnon.com/dnaman.html | N/A |
| MaxQuant | https://www.maxquant.org/mqlive/ | N/A |
| **Other** | | |

| Reagent/Resource | Reference or Source | Identifier or Catalog Number |
|---|---|---|
| 0.2 μm PVDF membrane | Millipore | ISEQ00010 |
| 0.22 μm PVDF Filter | Millipore | SLGVR33RS |
| Protein A/G magnetic beads | Selleck | B23202 |
| BCA Protein Quantification Kit | Vazyme | E112-01 |
| Reverse Transcriptase Mix | Vazyme | R222-01 |
| SYBR qPCR Mix | Vazyme | Q511-02 |
| Cell Counting Kit 8 | Abcam | ab228554 |
| Alanine aminotransferase Assay Kit | NanJing JianCheng | C009-2-1 |
| Aspartate aminotransferase Assay Kit | NanJing JianCheng | C010-2-1 |
| Blotting Electrophoresis Cell | Tanon | VE-180 |
| real-time qPCR platform | Roche | LightCycler 480 II |
| ZORBAX Eclipse Plus C18 columns | Agilent | 959758-902 |
| Triple Quadrupole LC/MS | Agilent | 1290 infinity II/6470 |

## Preparation of in slico drug screening and visualized *csa*-DEHP details

### Drug screening

To identify TCMP and chemical component library of CCLCM containing chemical compositions that aid in detoxification and toxin elimination, a Venn analysis was performed to identify effective TCM materials with a frequency of occurrence equal to or greater than 10. To identify compounds that have the potential to reduce DEHP accumulation, candidate compounds obtained from systematic pharmacology of the TCMSP (http://lsp.nwu.edu.cn/tcmsp.php), were evaluated based on their OB, DL, and half-life. TCMSP serves as a premier resource for in-depth pharmacological studies on TCM, bolstered by its extensive and meticulously curated database. Spanning across its expansive array of content, there are 499 unique Chinese herbs, 29,384 individual components, 3311 prospective therapeutic targets, and a catalog of 837 conditions intricately associated with these entities. By leveraging TCMSP, investigators can delve into the intricacies of TCM's holistic interactions, facilitating a comprehensive comprehension of its underlying mechanisms of action and potential biomarkers for diverse ailments (Ru et al, 2014). A pivotal aspect of TCM's pharmacological scrutiny lies in its assessment of OB, a metric symbolizing the proportion of the drug that successfully enters the systemic circulation. TCMSP plays a pivotal role in unraveling this critical parameter, as it facilitates the prediction of drug absorption kinetics, reflecting the efficacy and convenience of administration (Xu et al, 2012). Enhanced OB typically denotes improved therapeutic efficacy and ease of use, thereby enhancing patient compliance and overall medication success. DL, as a central concept in pharmaceutical

innovation, appraises the probability that a small molecular entity would possess optimal pharmacological characteristics. By integrating this evaluation into TCMSP's data, we can refine our compound selection and generate more promising lead compounds for TCM research, fostering a more rigorous scientific foundation and translating theoretical discoveries into tangible therapeutic advancements (Tao et al, 2013; Wei et al, 2020). Thus, the synergistic integration of TCMSP's database with DL assessments elevates the rigor of TCM therapeutics, translating intricate concepts into practical clinical applications. In our study, the criteria for screening candidate compounds were set as follows in the TCMSP system: OB ≥ 30%, DL ≥ 0.18, and half-life ≥4 h. Subsequently, topological calculations were performed on TCM and common effective chemical components, and candidate drugs decreasing DEHP accumulation were identified by screening for effective components with a degree of ≥3.

## Screening of ssDNA aptamers specifically targeting DEHP

A ssDNA library was utilized to screen the potential aptamers. This library possessed a 40-mer randomized region and fixed 20-mer primer binding sequences at both ends, guaranteeing effective amplification using fixed primers (Appendix Table S5). The ssDNA library was then immobilized onto magnetic streptavidin beads through hybridization with a complementary strand P3 (5′-TCAAGAGGTAGACGC-3′), which was complementary to a random ssDNA library primer region and labeled with biotin. DEHP was added to the solution containing the immobilized ssDNA library on magnetic beads and incubated to facilitate dissociation of the ssDNA-DEHP complex from the magnetic beads. The supernatant, which contained the ssDNA-DEHP complex, was used as the template for PCR amplification, followed by agarose gel electrophoresis, recovery, and purification. The PCR conditions were as follows: denaturation at 95 °C for 5 min, followed by denaturation at 95 °C for 30 s, annealing at 59 °C for 30 s, extension at 72 °C for 1 min, repeated for 30 cycles, and finally extension at 72 °C for 10 min. In the process of enriching the ssDNA library, the invention improved the screening efficiency by reducing the library concentration, shortening the incubation time, and reverse-screening and separating the nucleic acid aptamer sequence specifically bound to DEHP. In this study, we calculated the proportion of ssDNA recovered to the ssDNA input to obtain the retention rate of each round. At the same time, with the increase in the number of screening rounds, the ssDNA that failed to bind with DEHP or had weak binding ability was continuously removed, and the ssDNA that could bind with DEHP with high affinity was gradually enriched (Appendix Table S6). The screening process was ended, when the retention rate did not change significantly. Subsequently, the enriched ssDNA library was prepared, followed by clone sequencing, sequence analysis, and affinity and specificity experiments.

## Sequencing and characteristics of the aptamers

The final ssDNA library was amplified by using PCR, and the resulting amplified products were ligated to the pMD18-T plasmid and transformed into *Escherichia coli*. Next, 50 individual clones were randomly selected for further sequencing. The homology of these sequences was analyzed using DNAMAN software (Version 6.0.3.99, California, USA). The Gibbs free energy (△G) and secondary

structure predictions of the sequences were analyzed using beneficial RNA Structure software (Version 6.4, Rochester, New York, USA). After homology analysis, secondary structure prediction, and free energy calculation, the candidate aptamers against DEHP were ultimately determined. 5′-Carboxyfluorescein (FAM) was labeled for subsequent analyses of affinity and specificity. Note that the FAM probe was labeled on the aptamers using methods according to a previous study (Dadmehr et al, 2023). We chose FAM as the fluorescent marker to test the affinity and specificity of the aptamers, as FAM is preferred for its intense fluorescence emission, particularly in dilute solutions (Lu et al, 2015; Ma et al, 2016).

## Validation of the affinity and specificity of the aptamers targeting DEHP

The affinity and specificity of the candidate sequences were verified by the graphene oxide fluorescence method, and aptamers with high affinity and high specificity for recognizing DEHP were selected. For optimal performance, the ratio of graphene oxide to the candidate aptamers was optimized as a crucial factor in the detection process. Initially, FAM-labeled candidate aptamers underwent the same heating and cooling treatment as in the selection process (predenaturation at 95 °C for 10 min, followed by an ice bath for 10 min and then room temperature for 10 min). The FAM-labeled candidate aptamers were incubated with graphene oxide at different mass ratios of ssDNA to graphene oxide (1:0, 1:5, 1:10, 1:25, 1:50, 1:100, 1:150, 1:200). After a 30-min incubation at 37 °C with light shading and oscillation, the solution was centrifuged at 4 °C and 14,000 rpm for 15 min. The fluorescence intensity of the supernatant (emission wavelength 520 nm, excitation wavelength 494 nm) was measured to determine the optimal adsorption mass ratio, which was found to be a mass ratio of graphene oxide to ssDNA of 100:1.

To examine the affinity, the FAM-labeled candidate aptamers underwent the same heating and cooling treatment as the selection process. Subsequently, DEHP solutions with various concentrations were shaken and incubated at 37 °C in the dark for 1.5 h in the presence of the aptamers. Graphene oxide (mass ratio of graphene oxide to ssDNA = 100:1) was added to various reaction solutions and shaken in the dark for 30 min. After incubation, the solution mentioned above was centrifuged at 14,000 rpm for 15 min, and the supernatant was collected for fluorescence intensity measurement. The binding curves between fluorescence intensity and different concentrations of incubated DEHP were obtained using GraphPad Prism 9 (GraphPad, San Diego, CA). The dissociation constant ($Kd$) values were estimated through nonlinear regression analysis.

To assess specificity, the FAM-labeled candidate aptamers were mixed and incubated with either the positive target (DEHP) or the negative targets (dibutyl Phthalate (DBP), di-n-octylo-phthalate (DNOP), diethyl phthalate (DEP)) at 37 °C in the dark for 1.5 h. A control without a target was conducted simultaneously. Each experiment was repeated three times, and the error bars represent the standard deviation of these repeated measurements. Graphene oxide (mass ratio of graphene oxide to ssDNA = 100:1) was added to various reaction solutions and shaken in the dark for 30 min. After incubation, the solution was centrifuged at 14,000 rpm for 15 min, and the fluorescence intensity of the supernatant was measured to determine the specific recognition ability of different aptamers to DEHP.

## 3D structure analysis of the aptamer targeting DEHP

To evaluate the interaction between the best aptamer (Seq.10) and the DEHP molecule, the three-dimensional structures of both entities were modeled. To accomplish this, several types of software and websites were utilized, such as RNA Structure and the 3dRNA-2.0 website (http://biophy.hust.edu.cn/new/3dRNA), for aptamer modeling. In addition, the PubChem website (https://pubchem.ncbi.nlm.nih.gov/) was utilized to obtain the three-dimensional structure of DEHP. Furthermore, the molecular docking software AutoDock Tools (Version 1.5.6, California, USA) was employed for the molecular docking of the aptamer and DEHP.

## Establishment of visualized *csa*-DEHP

The synthesis of the visualized *csa*-DEHP triblock copolymers was carried out using a previously reported method with slight modifications. In brief, Cy5-labeled aptamer copolymers were initially synthesized. 10 μM Aptamer-COOH (15 μL) reacted with 1-ethyl-3-(3-dimethyl aminopropyl) carbodiimide (EDC, 0.2 mM, 50 μL) and N-hydroxysuccinimide (NHS, 0.2 mM, 50 μL) under stirring at room temperature for 10 min to form activated Aptamer-COOH (Aptamer-COOH/NHS/EDC molar ratio of 15:1000:1000), followed by addition of Cy5 probe (10 μM, 1.5 μL). The reaction was stirred at 60 rpm for 4 h at 4 °C. Subsequently, the reaction mixture was dialyzed against deionized water for 3 days using a dialysis membrane (molecular cutoff: 4000 MW) to eliminate the unreacted Cy5 probe. The resulting supernatant was freeze-dried to yield a white flocculated Cy5-labeled aptamer (covalent binding). To conjugate DEHP with the Cy5-labeled aptamer, the aptamer was dissolved in 100 μL of deionized water. Subsequently, DEHP (10 mM, 100 μL) was added to the solution at 4 °C. The reaction was allowed to proceed overnight with gentle stirring. To remove any unreacted components, the reaction solution was dialyzed against deionized water (molecular cutoff: 4000 MW) for 3 days to eliminate the unreacted DEHP. The resulting supernatant was freeze-dried to obtain DEHP-aptamer-CONH$_2$-Cy5 (*csa*-DEHP). The aptamer interacted with DEHP through hydrogen bonding. The resulting polymer was characterized using $^1$H NMR (JEOL 400 MHz, performed by Public technology platform, Nanjing Medical University) and fluorescence spectroscopy (Excitation wavelength: 650 nm, Emission wavelength range: 500–800 nm, Excitation slit width: 2.5 nm, Emission slit width: 10 nm, Scan rate:1200 nm/min, Average time: 0.5 s, Emission data interval: 1 nm).

For drug screening and efficacy of targeting DEHP, mouse PHs was exposed to 150 nM *csa*-DEHP for 12 h, followed by with indicated drugs according to time points and concentrations in the absence of *csa*-DEHP. Fluorescent images were observed with CLSM (LSM800, Zeiss, Germany) and processed using the ZEN 2012 (blue edition) imaging software. The normalization of the fluorescence signals was calculated by the fold change of the red fluorescent intensity from each group compared to their corresponding controls (set as "1").

## Immunocytochemistry analysis

Mouse PHs were treated with 150 nM *csa*-DEHP. Following a 12-h incubation, the cells were treated with Golgi-Tracker Green solution (diluted 1:100, Beyotime, Shanghai, China) for 30 min at 4 °C. After rinsing with PBS, fresh medium was added to the cells, and they were incubated for an additional 30 min at 37 °C. Subsequently, nuclear staining with DAPI was carried out, and the signals were visualized using a laser confocal microscope, capturing images from nonoverlapping fields. DEHP was labeled with a red marker, while the Golgi complex was visualized in green. For tracing the *csa*-DEHP in other organelles, ER-Tracker Green solution (diluted 1:1000, Beyotime, Shanghai, China), Mito-Tracker Green solution (diluted 1:10,000, Beyotime, Shanghai, China), and Lyso-Tracker Green solution (diluted 1:10,000, Beyotime, Shanghai, China) were preheated to 37 °C. The cells were treated with the respective tracker solutions for 30 min at 37 °C. Following a PBS wash, the cells were incubated with fresh culture medium for an additional 15 min at 37 °C. Fluorescent images were observed with CLSM (LSM800, Zeiss, Germany) and processed using the ZEN 2012 (blue edition) imaging software.

## Preparation of experimental models and subject details

### Cell culture

Mouse PHs were prepared from male C57BL/6 mice via the collagenase IV (Gibco, New York, USA) perfusion method as described previously and cultured in DMEM with high glucose (Gibco, New York, USA) at 37 °C and 5% CO$_2$.

## Cell viability assay

A CCK-8 toxicity assay was performed to assess the potential toxic effects of small molecule compounds that may eliminate the detrimental impact of DEHP from mouse PHs. Briefly, $10^4$ cells were seeded into each well of a 96-well plate and incubated at 37 °C overnight. After synchronization with serum-free DMEM, cells were transferred into 100 μL of serum-free DMEM containing either small molecule compounds or vehicles (0.1% DMSO) and incubated for an additional 24 h. Next, 10 μL WST-8 reagent (Jiancheng, Nanjing, China) was added to each well and incubated at 37 °C for 2 h. Finally, the absorbance at 450 nm was measured using a microplate reader.

## Cell transfection

The plasmids encoding mouse Uroc1 and its mutants were synthesized by Tsingke (Beijing, China). All transient transfections were conducted using Lipofectamine 3000 (Invitrogen, Carlsbad, CA, USA) according to the manufacturer's instructions. For Uroc1 knockdown, shRNA targeting mouse Uroc1 were designed, validated, and synthesized by GenePharma (Shanghai, China). Detailed shRNA sequences were: 5′-CACCGCCCAAGAGAAGTT CTTCTTCTTTCAAGAGAAGAAGAAGAACTTCTCTTGGGTTT TTTG-3′ for Uroc1; 5′-CACCCTTCTCCGAACGTGTCACGTT TCA AGAGAACGTGACACGTTCGGAGAACTTTTTTG-3′ for negative controls (NC). The data were representative of at least four independent experiments.

## Ethics statement

All animal procedures in this investigation conformed to the Guide for the Care and Use of Laboratory Animals published by the US

 

National Institutes of Health (NIH publication No. 85–23, revised 1996) and the approved regulations set by the Laboratory Animal Care Committee at China Pharmaceutical University (Approval number SYXK-2021-0011).

## Animals

Male C57BL/6J mice were purchased from GemPharmatech (Nanjing, China) and were maintained in a 12 h light/12 h dark cycle (light/dark, 12:12) in a temperature- and humidity-controlled environment. To evaluate the effects of luteolin or trans-UCA on reducing DEHP accumulation from mouse livers, 8-week-old male mice weighting 20–23 g were randomly divided into three groups, namely, the CTL group, DEHP group, and DEHP+luteolin group ($n = 6$ per group). Mice in the DEHP group received daily intragastric (i.g.) administration of DEHP (10 mg/kg body weight/day) for 28 days. Mice in the DEHP+luteolin group received daily i.g. administration of DEHP for 14 days. At this time point, these mice then received daily i.g. administration of luteolin (10 mg/kg body weight/day) for 14 days in the presence of DEHP. Mice in the CTL group received an equal volume of corn oil solution administered in the same way for 28 days. The luteolin used in this experiment was purchased from J&K Scientific Technology Co., Ltd. (Beijing, China). The dose of luteolin was selected according to a previous study reporting that luteolin, with a maximum dose regimen reaching up to 100 mg/kg body weight per day, ameliorates LPS-induced liver injury in mice (Wang et al, 2021a), indicating that the dose of luteolin used in our study exhibited high biocompatibility and less hepatic/renal toxicity.

For liver-specific overexpression of Uroc1, we transduced a single-stranded adenoviral-associated virus 8 (AAV8) system carrying either Uroc1 CDS (accession number NM_001347329.1) or green fluorescent protein (GFP) into mice at a dose of $1 \times 10^{12}$ vg through tail vein injection under the hepatocyte-specific thyroid binding globulin (TBG) promoter. The dose of AAV was chosen based on a previous study showing that this dose functionally manipulates gene expression in mouse hepatocytes (Chen et al, 2019). After 1 month, these mice were randomly divided into three groups, namely, the AAV-TBG-DEHP group, AAV-TBG-DEHP +luteolin group, and AAV-Uroc1-DEHP+luteolin group ($n = 6$ per group). Mice in the AAV-TBG-DEHP group received daily i.g. administration of DEHP (10 mg/kg body weight/day) for 28 days. Mice in the AAV-TBG-DEHP+luteolin and AAV-Uroc1-DEHP +luteolin groups received daily i.g. administration of DEHP for 14 days. At this time point, these mice then received daily i.g. administration of luteolin (10 mg/kg body weight/day) for 14 days in the presence of DEHP.

To further verify the effect of trans-UCA on the reduction of DEHP in mice, 8-week-old male mice were randomly divided into three groups, namely, the DEHP group, and DEHP+trans-UCA group ($n = 6$ per group). Mice in the DEHP group received daily i.g. administration of DEHP (10 mg/kg body weight/day) for 28 days. Mice in the DEHP+trans-UCA group received daily i.g. administration of DEHP for 14 days. At this time point, these mice then received daily i.g. administration of trans-UCA (10 mg/kg body weight/day) for 14 days in the presence of DEHP. The dose of trans-UCA was selected according to a previous study reporting that cis-UCA inhibited skin allograft rejection and acute graft-versus-host disease (Kaneko et al, 2009). Trans-UCA was dissolved

in normal saline and injected intraperitoneally. The trans-UCA used in this experiment was purchased from Beijing Solarbio Science & Technology Co., Ltd (Beijing, China).

At the end of each animal experiment, mouse urine was collected according to a reported method (Chew and Chua, 2003), and all mice were humanely killed for the collection of serum, liver, kidney and white adipose tissue samples. Despite being initially fixed for pathological evaluation, all samples were promptly quick-frozen using liquid nitrogen and stored at −80 °C for subsequent analysis.

## Serological analyses

Blood samples were collected in non-heparinized tubes and centrifuged at 4000 rpm for 10 min at 4 °C. Serum levels of AST, ALT, and glucose were determined by using commercial kits purchased from Jiancheng Institute of Biotechnology (Nanjing, Jiangsu, China). Serum levels of angiotensin was determined with an ELISA assay kit (COIBO BIO, Shanghai, China) according to manufacturer's instructions.

## Preparation of samples for HPLC-MS analysis

### Cell sample preparation for HPLC-MS analysis
Cell samples were processed as follows: harvested, washed thrice with PBS, then centrifuged at 14,000 rpm for 5 min. The supernatant was discarded, and cells were resuspended in 100 μL PBS. Next, they were mixed with 100 μL of DEHP-D4 (10 ng/mL), and 400 μL of methanol. The mixture underwent vortexing for 5 s, followed by ultrasound treatment (4 °C, 60% intensity, 10-s intervals, for a total of 3 min). The sample was vortexed again for 5 min, centrifuged at 14,000 rpm for 10 min, and ~80% of the supernatant was dried under nitrogen. The remaining residue was reconstituted in acetonitrile and subjected to another 10-min sonication. A final centrifugation at 14,000 rpm for 5 min ensued. The supernatant was discarded, and the resulting solution was redissolved in 200 μL of acetonitrile for HPLC-MS analysis.

## Animal sample preparation for HPLC-MS analysis

50 μL of biofluids, such as serum or urine, were collected in a 1.5 mL EP tube, and for tissue samples (like liver or white adipose tissues), a 50 mg aliquot was used. Prior to processing, 2 mL of methanol was added to each sample. Tissue samples underwent an additional step known as magnetic bead homogenization to obtain a homogenate solution. Following this, pre-diluted internal standards were included as well, with DEHP-D4 (10 ng/mL in methanol) and MEHP-D4 (10 ng/mL in methanol), after which the mixture was subjected to sonication for 10 min and centrifugation at 14,000 rpm at 4 °C for 10 min. To conduct HPLC-MS analysis, the supernatant (1.5 mL) was transferred to a new EP tube, the liquid was completely evaporated under a nitrogen stream, and then resuspended in 200 μL of acetonitrile.

## HPLC–MS analysis

The triple quadrupole LC/MS-MS analysis was performed using an Agilent 1290 infinity II/6470. The system control and data analyses were performed by MassHunter (the software version: Version

12.1). Chromatographic separation was achieved using a Ecl and Plus C18 columns (100 mm × 2.1 mm × 1.8 μm) with a guard column (Agilent Technologies, Palo Alto, CA, USA). The HPLC was operated with a gradient mobile phase system consisting of water (phase A) and methanol contained 10% ammonium acetate (5.0 mM) (phase B) at gradient elution. Detailed information of gradient elution was presented in Appendix Table S7. A 20 μL sample was injected into the system with the auto-sampler conditioned at 4 °C. The mass spectrometer was operated in both positive and negative ion mode. MRM transitions were performed at $m/z$ 391 → 149, $m/z$ 391 → 167 for DEHP, at $m/z$ 395 → 153 for DEHP-D4, $m/z$ 277 → 127, $m/z$ 277 → 134 for MEHP, $m/z$ 281 → 138 for MEHP-D4. Detailed information for MS parameters were listed in Appendix Table S8 and Appendix Table S9.

## Label-free quantification and protein identification

To identify candidate proteins involved in the reduction of DEHP by luteolin/*trans*-UCA, label-free proteomic analysis was performed on mouse PHs. The cells were initially treated with 10 μM DEHP for 12 h, followed by treatment with either 10 μM luteolin (or 10 mM *trans*-UCA) or 0.1% DMSO for an additional 12 h in the absence of DEHP. A detailed proteomics analysis was conducted following standard protein digestion and LC–MS/MS analyses by Shanghai Applied Protein Technology Co. in Shanghai, China. The resulting MS raw data for each sample were combined and analyzed for identification and quantitation using MaxQuant software (Version 1.5.3.17). $P < 0.05$ and fold change ≥1.5 (for luteolin-treated group) or ≥2 (for *trans*-UCA-treated group) were considered to be statistically difference. $P < 0.05$ was considered as a threshold of GO enrichment analysis. Note that FDR correction was not used for proteomic data analysis.

## GO annotation

To investigate the differential expressed proteins, we initially performed a local search for protein sequences using the NCBI BLAST+client software (ncbi-blast-2.2.28±win32.exe), seeking homologous sequences across the database. Subsequently, Inter-ProScan was employed to augment and refine the annotation by examining protein domains and conserved functional regions. The sequences were then subjected to GO annotation through Blast2GO, a bioinformatics tool that assigns biological process to the proteins. The obtained GO annotations were systematically analyzed and visualized, utilizing R scripts to create graphical representations of the data. In addition, the GO functional annotations of all differentially expressed proteins were compared with those of the reference species (or all experimentally identified proteins). A Fisher's Exact Test was performed to determine the significance of the differences and identify enriched functional categories (with a $P$ value < 0.05).

## Molecular docking analysis

The 3D structures of candidate targets were obtained from the PDB database (http://www.rcsb.org/) in PDB format, specifically selecting "Mouse sapiens only" as the organism. The 3D conformers of candidate compounds were acquired from the PubChem database (https://pubchem.ncbi.nlm.nih.gov/) in SDF format. Subsequently,

the structures were imported into the Molecular Operating Environment (MOE) to obtain the docking score. A docking score with a higher absolute value indicates a better result.

## Reverse transcription-quantitative real-time PCR (RT-qPCR)

Total RNA was extracted from liver tissues and PHs using TRIzol reagents (Invitrogen, Carlsbad, CA, USA). cDNA was synthesized from the total RNA using a reverse transcription kit (Takara, Dalian, China). Gene expressions were measured using qPCR (LightCycler 96, Roche, USA) with SYBR green kits (Vazyme, Nanjing, Jiangsu, China). Mouse 36B4 was used as the normalization control. A complete list of the primers was showed in Appendix Table S10.

## Western blotting analysis

For protein analysis, the harvested cells or liver tissue homogenates were suspended in RIPA buffer containing a protease inhibitor cocktail (Roche, USA) and PMSF (Beyotime, Shanghai, China). Equal amounts of protein were loaded and separated by 10% SDS/PAGE, and then transferred onto a polyvinylidene difluoride membrane (Millipore, USA). The samples were blotted overnight using appropriate antibodies against mouse Uroc1 (Cat. No. 25032-1-AP; 1:1000 dilution, Proteintech, Chicago, USA), phospho-ERK1/2 (Cat. No.9102S; 1:1000 dilution, Cell Signaling, Boston, USA), total ERK1/2 (Cat. No.4377S; 1:1000 dilution, Cell Signaling, Boston, USA), N-Cadherin (Cat. #13116, 1:1000 dilution, Cell Signaling, Boston, USA), Vimentin (Cat. #5471; 1:1000 dilution, Cell Signaling, Boston, USA), α-SMA (Cat. No. 14395-1-AP; 1:1000 dilution, Proteintech, Chicago, USA) or β-Actin (Cat. No. 81115-1-RR; 1:4000 dilution, Cell Signaling, Chicago, USA). These experiments were performed in triplicate.

## Protein half-life and ubiquitination analyses

The half-life of the Uroc1 protein was determined using the cycloheximide (CHX) chase assay. Mouse PHs were treated with 10 μM DEHP for 12 h and luteolin for 12 h. After 24 h, DMEM containing 10 μg/mL CHX (Sigma-Aldrich, St. Louis, MO, USA) was added. Cells were harvested at the specified time points and subjected to western blot analysis. Ubiquitination of the Uroc1 protein was measured by a co-IP assay. Mouse PHs were similarly treated with 10 μM DEHP for 12 h, followed by a single stimulation with 10 μM luteolin for 6 h in the absence of DEHP. Subsequently, these cells were further treated with a combination of 10 μM luteolin and 10 μM MG132 (Sigma-Aldrich, St. Louis, MO, USA) or 20 nM bafilomycin A1 (Baf A1) (MCE, New Jersey, USA) for 6 h. Cells were lysed, and lysates were precipitated with anti-Uroc1 antibody and precleared protein A/G PLUS-Agarose beads (Roche, Basal, Switzerland). Ubiquitinated proteins within the IP were detected by western blot using a monoclonal anti-ubiquitin antibody (Cat. No. 10201-2-AP; 1:500 dilution, Proteintech, Chicago, USA).

## Statistical analysis

Statistical analysis was performed by using GraphPad Prism 9 (GraphPad, San Diego, CA). Groups of data are presented as the

## The paper explained

### Problem

Di-(2-ethylhexyl) phthalate (DEHP), widely detected in both environmental and clinical samples, poses a severe threat to endocrine homeostasis. However, no efficacious therapy exists specifically targeting DEHP excretion from the liver to mitigate its harmful effects on the endocrine system.

### Results

Conducting extensive network pharmacology analyses and utilizing the highly efficient visualized *csa*-DEHP platform, we have identified luteolin as a potent therapeutic agent for efficient detoxification of DEHP in the mouse liver. Mechanistically, luteolin accelerates the protein degradation of hepatic urocanate hydratase 1 (Uroc1). In addition, *trans*-UCA, as the substrate of Uroc1, inhibits the activation of ERK1/2 signaling cascade, thereby facilitating the lysosomal exocytosis of the toxic compound. Ultimately, luteolin achieves detoxification of DEHP in the mouse liver by modulating the activity of the Uroc1/UCA axis.

### Impact

These results provide strong implications for harnessing the therapeutic potential of compounds like luteolin and its endogenously linked *trans*-UCA, in combating ailments stemming from environmental pollutants such as DEHP and its analogs. Their demonstrated detoxification capabilities will shed lights on the clinical treatment of pollution-induced disorders.

mean $\pm$ SD (standard deviation). The paired Student's *t* test (two-tailed) and one-way ANOVA followed by Fisher's least significant difference (LSD) post hoc test were performed to analyze the data, where appropriate. A value of $P < 0.05$ was considered statistically significant.

## Data availability

The data supporting the findings are available within the article and Supporting Information. The mass spectrometry proteomics data have been deposited to the ProteomeXchange Consortium (https://www.ebi.ac.uk/pride/) via the PRIDE partner repository with the dataset identifier PXD055165 for Luteolin and PXD055187 for Trans-UCA.

The source data of this paper are collected in the following database record: biostudies:S-SCDT-10_1038-S44321-024-00160-9.

## Peer review information

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

## Acknowledgements

This work was financially supported by grants from the National Key R&D Program of China (no. 2022YFA0807200 to SC, no. 2021YFF0702000 to CL), the National Natural Science Foundation of China (no. 32471201 to SC, no.

82470930 to CL), the Natural Science Foundation of Jiangsu Province (no. BK20220151 to SC), the Natural Science Foundation of Xinjiang Uygur Autonomous Region (no. 2023D01D09, no. 2023D01D05 to CL), the Natural Science Foundation of Chongqing, China (CSTB2023NSCQ-MSX0094 to CL), the Priority Academic Program Development of Jiangsu Higher Education Institutions (PAPD, to SC, CL, and WZ).

## Author contributions

**Huiting Wang**: Conceptualization; Formal analysis; Investigation; Visualization; Methodology; Writing—original draft; Project administration. **Ziting zhao**: Formal analysis; Investigation; Visualization. **Mingming Song**: Software; Formal analysis; Methodology. **Wenxiang Zhang**: Funding acquisition; Writing—review and editing. **Chang Liu**: Resources; Funding acquisition; Writing—review and editing. **Siyu Chen**: Conceptualization; Supervision; Funding acquisition; Project administration; Writing—review and editing.

Source data underlying figure panels in this paper may have individual authorship assigned. Where available, figure panel/source data authorship is listed in the following database record: biostudies:S-SCDT-10_1038-S44321-024-00160-9.

## Disclosure and competing interests statement

The authors declare no competing interests.

# Expanded View Figures

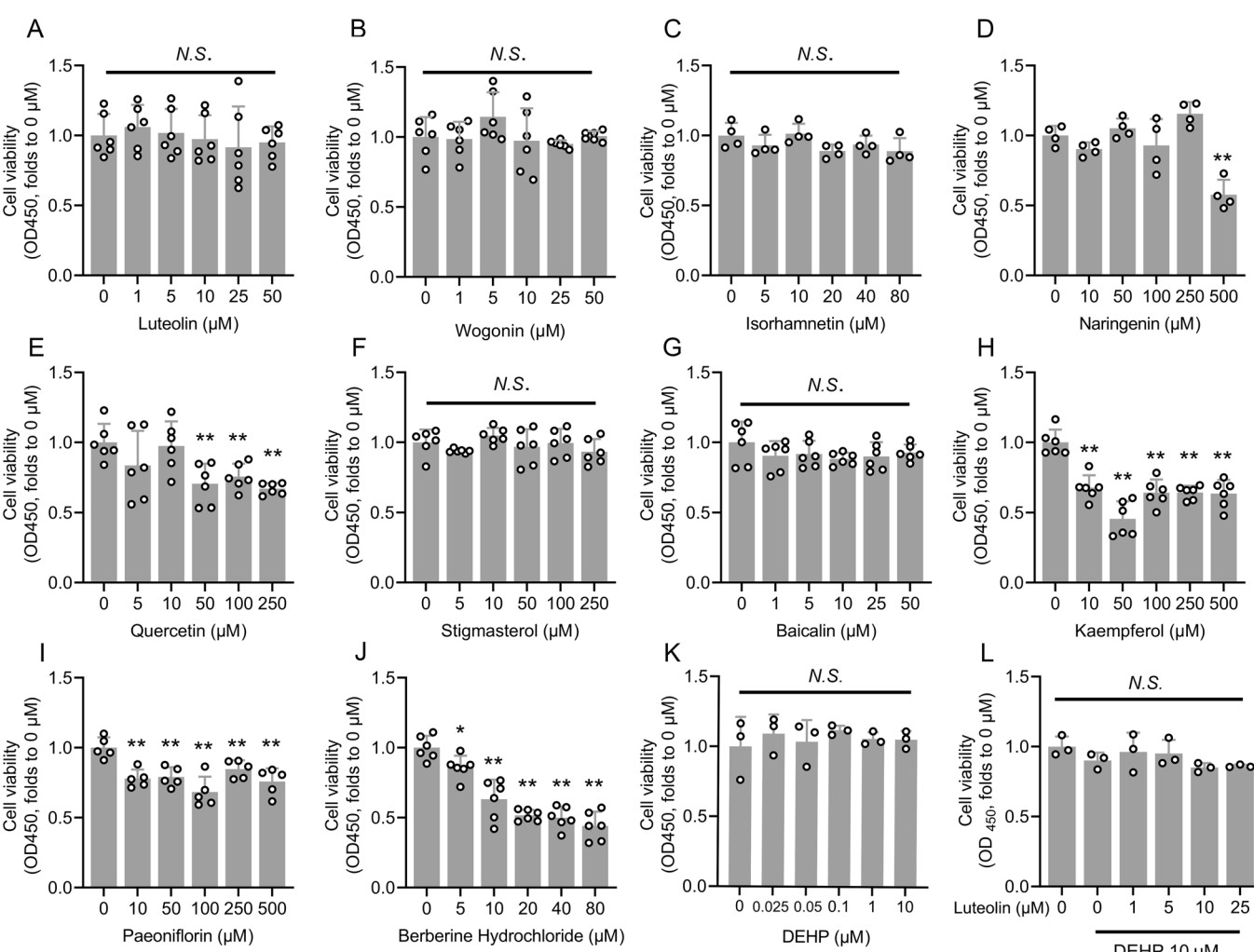

**Figure EV1. Cell toxicity analysis of candidate drugs in moue PHs.**

(A) Cell viability analysis of mouse PHs treated with indicated doses of luteolin. $n = 6$. (B) Cell viability analysis of mouse PHs treated with indicated doses of wogonin. $n = 6$. (C) Cell viability analysis of mouse PHs treated with indicated doses of isorhamnetin. $n = 4$. (D) Cell viability analysis of mouse PHs treated with indicated doses of naringenin. $n = 4$. (E) Cell viability analysis of mouse PHs treated with indicated doses of quercetin. $n = 6$. (F) Cell viability analysis of mouse PHs treated with indicated doses of stigmasterol. $n = 6$. (G) Cell viability analysis of mouse PHs treated with indicated doses of baicalin. $n = 6$. (H) Cell viability analysis of mouse PHs treated with indicated doses of kaempferol. $n = 6$. (I) Cell viability analysis of mouse PHs treated with indicated doses of paeoniflorin. $n = 5$. (J) Cell viability analysis of mouse PHs treated with indicated doses of berberine hydrochloride. $n = 6$. (K) Cell viability analysis of mouse PHs treated with indicated doses of DEHP. $n = 3$. (L) Cell viability analysis of mouse PHs exposed to 10 μM DEHP and then incubated with indicated doses of luteolin for another 24 h. $n = 3$. N.S., no significance. **$P < 0.01$ vs. 0 μM group. All the data were represented as the mean ± SD. The paired Student's $t$-test was employed to compare between two groups. Exact $P$ values are listed in Appendix Table S11.

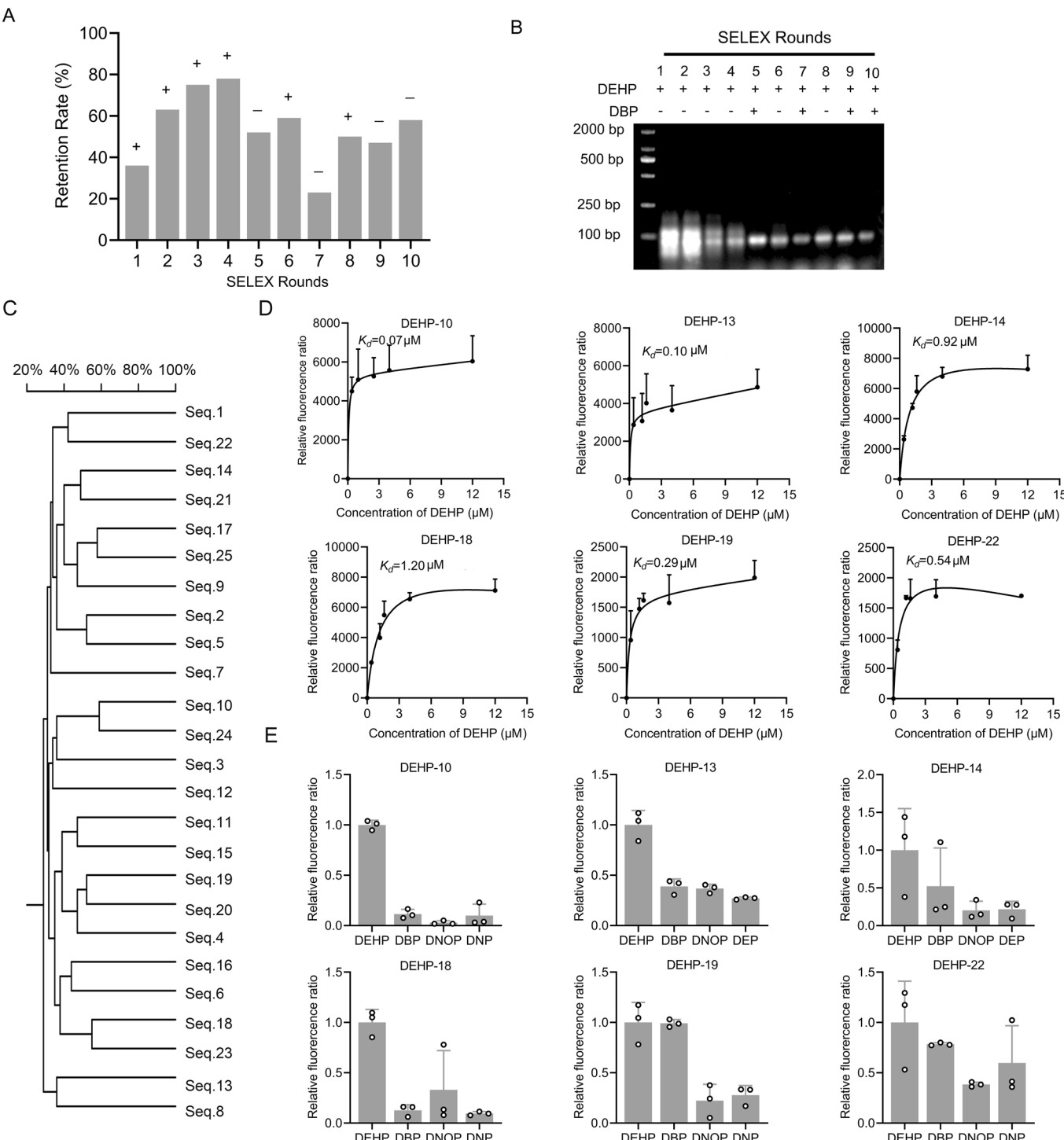

**Figure EV2.** **Screening of DNA-aptamer targeting the DEHP.**

(**A**) Retention rate of ssDNA sequences collected during the selection of DEHP aptamers using Capture-SELEX technology. (**B**) Electrophoretic verification of the ssDNA library from each screening round. (**C**) Phylogenetic tree analysis of candidate aptamers targeting DEHP. (**D**) Graphene oxide fluorescence assay for the affinity of candidate aptamers targeting DEHP. Data are fitted to a Michaelis-Menten model (curve) to calculate the dissociation constants. (**E**) Binding specificity of candidate aptamers to DEHP. $n = 3$. '$n$' represents technical replicates. All the data were represented as the mean ± SD.

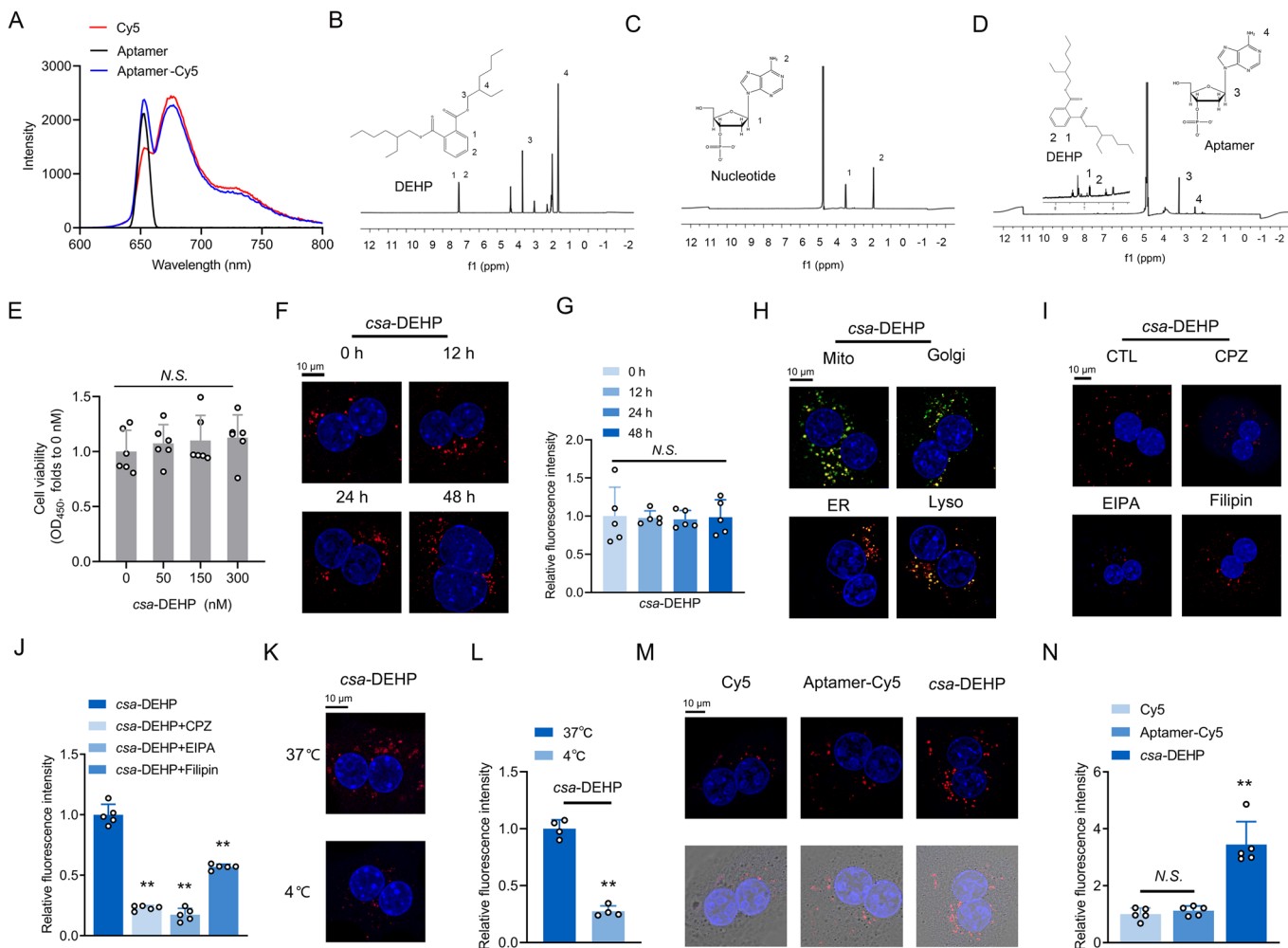

**Figure EV3. Validation of DNA-aptamer targeting the DEHP.**

(A) Fluorescent spectra of the Cy5, aptamer and aptamer-Cy5. (B) $^1$H NMR spectra of DEHP. (C) $^1$H NMR spectra of aptamer. (D) $^1$H NMR spectra of aptamer-DEHP. (E) A CCK-8 analysis for cell toxicity of *csa*-DEHP in mouse PHs that were treated with indicated doses of *csa*-DEHP for 24 h. $n = 6$. *N.S.*, no significance. (F) CLMS analysis of retention time of internal *csa*-DEHP in mouse PHs following incubation with 150 nM *csa*-DEHP for 12 h. (G) Quantitative analysis of the relative fluorescence intensity in (F). *N.S.*, no significance. $n = 5$. '$n$' represents biological replicates. (H) Organelle localization of *csa*-DEHP in mouse PHs treated with 150 nM *csa*-DEHP for 12 h, and subsequently stained with specific organelle fluorescent trackers. (I) Representative CLMS images of mouse PHs treated with either *csa*-DEHP alone or in combination with indicated inhibitors. (J) Quantitative analysis of the relative fluorescence intensity in (I). **$P < 0.01$ *vs.* *csa*-DEHP group, $n = 5$. '$n$' represents biological replicates. (K) Representative CLMS images of *csa*-DEHP in mouse PHs treated with 150 nM *csa*-DEHP for 1 h in either 37 °C or 4 °C. (L) Quantitative analysis of the relative fluorescence intensity in (K). **$P < 0.01$ *vs.* 37 °C group, $n = 4$. '$n$' represents biological replicates. (M) Representative CLMS images of mouse PHs treated with 150 nM Cy5, Aptamer-Cy5 and *csa*-DEHP, respectively, for 12 h. (N) Quantitative analysis of the relative fluorescence intensity in (M). **$P < 0.01$ *vs.* Cy5 group, $n = 5$. '$n$' represents biological replicates. Scale bar: 10 μm. Mito, Mitochondria. ER, Endoplasmic Reticulum. Golgi, Golgi apparatus. Lyso, Lysosome. Organelle, green. *csa*-DEHP, red. Nuclei, blue. All the data were represented as the mean ± SD. The paired Student's *t*-test was employed to compare between two groups. One-way ANOVA with a Fisher's LSD post hoc test was utilized to compare among multiple groups. Exact $P$ values are listed in Appendix Table S11.

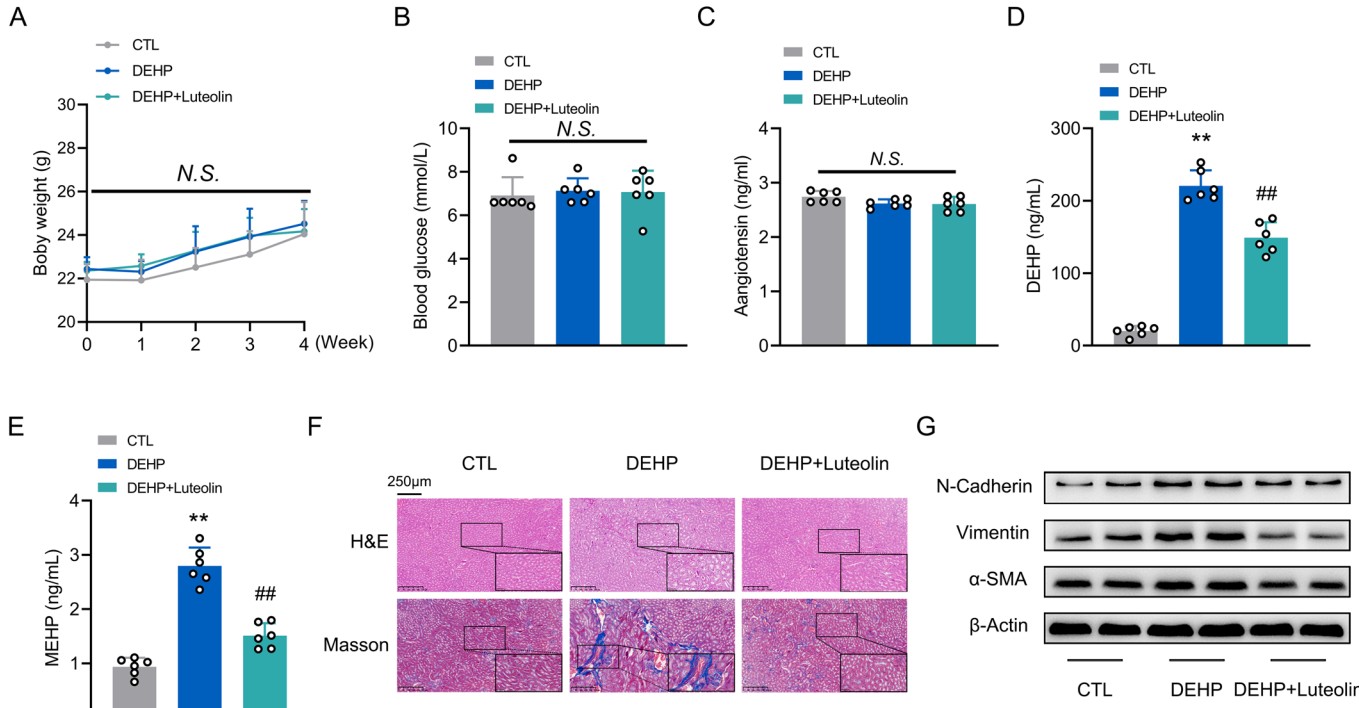

**Figure EV4.  The effects of luteolin on the metabolic profile of DEHP-treated mice.**

Mice subjected to a 14-day treatment with DEHP (10 mg/kg body weight/day), followed by an additional 14-day treatment with luteolin (10 mg/kg body weight/day) in the presence of DEHP. $n = 6$. (**A**) Body weight. *N.S.*, no significance. (**B**) Serum levels of glucose. *N.S.*, no significance. (**C**) Serum levels of angiotensin. *N.S.*, no significance. (**D**) HPLC-MS analysis of DEHP levels in the adipose tissue of mice. **$P < 0.01$ *vs.* CTL group. ##$P < 0.01$ *vs.* DEHP group. $n = 6$. (**E**) HPLC-MS analysis of MEHP levels in the adipose tissue of mice. **$P < 0.01$ *vs.* CTL group. ##$P < 0.01$ *vs.* DEHP group. $n = 6$. (**F**) The renal histopathological examination and collagen deposition were performed by H&E (up) and Masson's trichrome staining (down), respectively. (**G**) Western blot analyses of EMT and fibrotic markers expression in the kidney of mice. All the data were represented as the mean ± SD. The paired Student's *t*-test was employed to compare between two groups. One-way ANOVA with a Fisher's LSD post hoc test was utilized to compare among multiple groups. Exact *P* values are listed in Appendix Table S11.

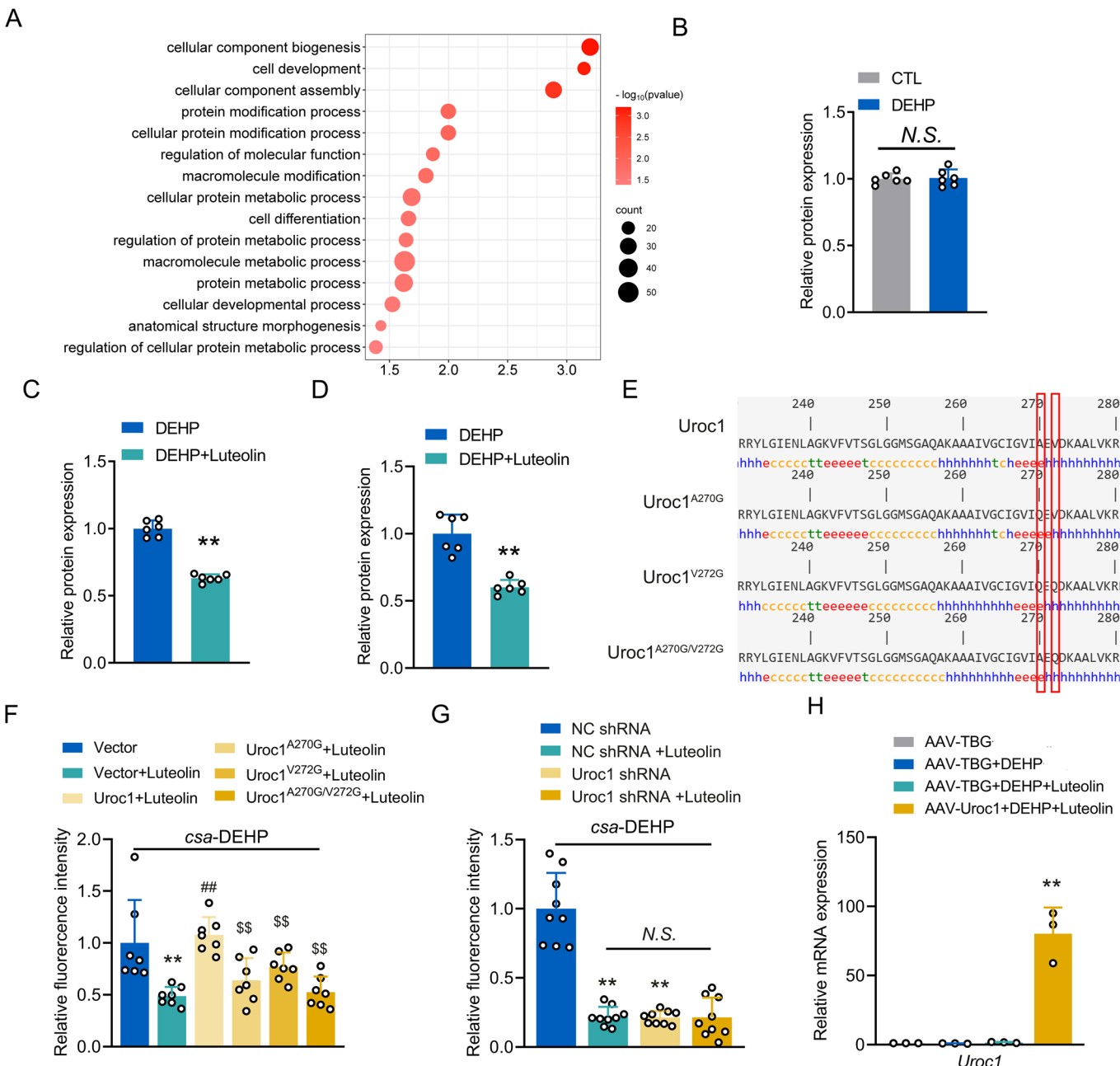

**Figure EV5. Uroc1 serves as the molecular target of luteolin.**

(A) GO enrichment analysis of differentially expressed proteins (biological process). The GO functional annotations of all differentially expressed proteins were compared with those of the reference species (or all experimentally identified proteins). A Fisher's Exact Test was performed to determine the significance of the differences and identify enriched functional categories (with a $P$ value < 0.05). (B) Quantitative analysis of protein expression in Fig. 3F. (C) Quantitative analysis of protein expression in Fig. 3G. **$P$ < 0.01 *vs.* DEHP group. $n = 6$. (D) Quantitative analysis of protein expression in Fig. 3H. *N.S.*, no significance. **$P$ < 0.01 *vs.* DEHP group. $n = 6$. '$n$' represents biological replicates. (E) A cluster of three evolutionarily conserved, positively charged amino acids in Uroc1 were mutated to Gln. Secondary structures of the wild-type and mutated mouse Uroc1 fragments were predicted using the SOPMA algorithm. h, helix. e, sheet. t, turn. c, coil. (F) Quantitative analysis of the relative fluorescence intensity in Fig. 4D. **$P$ < 0.01 *vs.* Vector group, ##$P$ < 0.01 *vs.* Vector+Luteolin group, $$$P$ < 0.01 *vs.* Uroc1+Luteolin group. $n = 7$. '$n$' represents biological replicates. (G) Quantitative analysis of the relative fluorescence intensity in Fig. 4E. *N.S.*, no significance. **$P$ < 0.01 *vs.* NC shRNA group. $n = 9$. '$n$' represents biological replicates. (H) RT-qPCR analysis of Uroc1 expression in the liver of mice treated as described in Fig. 3H. **$P$ < 0.01 *vs.* AAV-TBG + DEHP+Luteolin group, $n = 3$. All the data were represented as the mean ± SD. The paired Student's $t$-test was employed to compare between two groups. One-way ANOVA with a Fisher's LSD post hoc test was utilized to compare among multiple groups. Exact $P$ values are listed in Appendix Table S11.

