## [Peer Review File · EMBO Molecular Medicine]

Luteolin detoxifies DEHP and prevents liver injury by degrading Uroc1 protein in mice

Huiting Wang, Ziting zhao, Mingming Song, Wenxiang Zhang, Chang Liu, and Siyu Chen

Corresponding author(s): Siyu Chen (siyuchen@cpu.edu.cn) , Chang Liu (changliu@cpu.edu.cn), Wenxiang Zhang (wenxiangzhang@cpu.edu.cn)

Review Timeline:

Submission Date:	8th May 24
Editorial Decision:	7th Jun 24
Revision Received:	9th Sep 24
Editorial Decision:	1st Oct 24
Revision Received:	10th Oct 24
Accepted:	11th Oct 24

Editor: Zeljko Durdevic

Transaction Report:

7th Jun 2024

Dear Prof. Chen,

Thank you for the submission of your manuscript to EMBO Molecular Medicine. We have now received feedback from the two reviewers who agreed to evaluate your manuscript. Both referees recognize interest of the study but also raise important concerns that should be addressed in a major revision. Particular attention should be given to resolving methodological concerns. In addition, please provide better explanation of the selection process to identify drugs that can decrease DEHP accumulation and why the dose of 10 mg/kg of DEHP was chosen. If you would like to discuss further the points raised by the referees, I am available to do so via email or video. Let me know if you are interested in this option.

We would welcome the submission of a revised version within three months for further consideration. Please let us know if you require longer to complete the revision.

I look forward to receiving your revised manuscript.

Yours sincerely,

Zeljko Durdevic

We require:

- 1) A .docx formatted version of the manuscript text (including legends for main figures, EV figures and tables). Please make sure that the changes are highlighted to be clearly visible.
- 2) Individual production quality figure files as .eps, .tif, .jpg (one file per figure). For guidance, download the 'Figure Guide PDF': (<https://www.embopress.org/page/journal/17574684/authorguide#figureformat>).
- 3) A .docx formatted letter INCLUDING the reviewers' reports and your detailed point-by-point responses to their comments. As part of the EMBO Press transparent editorial process, the point-by-point response is part of the Review Process File (RPF), which will be published alongside your paper.
- 4) A complete author checklist, which you can download from our author guidelines (<https://www.embopress.org/page/journal/17574684/authorguide#submissionofrevisions>). Please insert information in the checklist that is also reflected in the manuscript. The completed author checklist will also be part of the RPF.
- 5) Please note that all corresponding authors are required to supply an ORCID ID for their name upon submission of a revised manuscript.

6) It is mandatory to include a 'Data Availability' section after the Materials and Methods. Before submitting your revision, primary datasets produced in this study need to be deposited in an appropriate public database, and the accession numbers and database listed under 'Data Availability'. Please remember to provide a reviewer password if the datasets are not yet public (see <https://www.embopress.org/page/journal/17574684/authorguide#dataavailability>).

13) Author contributions: You will be asked to provide CRediT (Contributor Role Taxonomy) terms in the submission system. These replace a narrative author contribution section in the manuscript.

14) A Conflict of Interest statement should be provided in the main text.

Please also suggest a striking image or visual abstract to illustrate your article as a PNG file 550 px wide x 300-800 px high.

**** Reviewer's comments ****

Referee #1 (Remarks for Author):

Wang et al. conducted an elegant study to explore which therapeutic agents and molecular targets might be useful in reducing DEHP accumulation in the liver. By developing a custom DNA-aptamer-based approach to track DEHP, they identified luteolin as a potential drug candidate. They report that luteolin enhances Uroc1 degradation (via targeting Ala270), promotes trans-UCA accumulation and thereby activates lysosomal exocytosis via ERK1/2 signaling, ultimately promoting DEHP release from hepatocytes and limiting the detrimental effects of phthalate exposure in the liver. The findings are interesting and potentially translational, even more so given the reported beneficial effects of luteolin in humans in terms of mortality risk (PMID: 36966325). However, some concerns, including methodological issues, should be addressed.

- Collateral organ damage, with a particular focus on the kidney: apart from the liver, the main target of phthalates is the kidney (see e.g. PMID: 38107267, 33736212, PMID: 34126474, PMID: 30462244, PMID: 24178978, PMID: 26100504). As reducing liver accumulation could lead to chronic over-exposure to DEHP/MEHP/MEHHP/MEOHP for the kidneys (PMID: 38107267), where they induce EMT and fibrosis (PMID: 29669060), and this could imply damage in terms of filtration rate or albuminuric loss, although lutein could be protective in this sense. As the authors aim to identify a potential anti-DEHP drug, they should also investigate the effect on the renal profile as described above (providing data on renal parameters and fibrosis at least).
- In addition to the previous comment, the metabolic phenotype of the mice should at least be examined in summary: were there any changes in body weight, glucose and blood pressure levels between ctrl, DEHP and DEHP+luteolin mice? Was there an over-accumulation of DEHP in adipose tissue?
- The authors mention in the text several times the adoption of untargeted proteomics (see results "Hepatic Uroc1 is responsive to luteolin treatment at the translational level" and "RK1/2-orchestrated lysosome pathway is involved in mediating the luteolin/trans-UCA-facilitated hepatic DEHP release" paragraphs, and the figures 3 and 6). Nevertheless, in the same figures, and in the correspondent text, it seems that the authors have rather adopted a transcriptomic screening (in the subsequent panels they refer to genes, not to proteins, e.g. "As shown in Fig. 3B, a cluster of 21 genes was changed in response to luteolin treatment"; "A total of 76 genes were altered by trans-UCA treatment, with 27 upregulated and 49 downregulated") which is not described in the methods (were they RNA-seq? custom/non-custom microarray screening?). Might the author clarify on this? Are the authors willingly to deposit the omics data in a public repository?
- The authors focused on intragastric absorption, and this should be better clarified in the manuscript (especially when discussing implications, limitations and discussion), as it is known that there are other major routes of absorption and that organ accumulation also depends on the route of absorption (e.g. PMID: 35565138), which slightly limits the generalisability of the authors' work in terms of actual treatment of DEHP exposure.
- Might the authors provide a GO analysis for luteolin modulated pathways in figure 3, as provided in figure 6 for trans-UCA to observe more easily the difference between the metabolite replenishment vs the drug approach? (e.g., Acaa1b has a opposite behaviour).
- Methods: the animal protocol states "These mice were treated with either DEHP or trans-UCA (10 mg/kg body weight/day) in a manner similar to the experiment evaluating the effect of luteolin." What is meant by "similar"? Was there any difference, however small? Could the author precisely report the study protocol for the trans-UCA?
- Statistical analysis: Was a correction for multiple testing correction (e.g. FDR) used for -omics results? This should be clarified in the main text (even if it was not used, it should be clearly reported in the manuscript).
- Figure 5: typo ("tran" instead of "trans"-UCA) in panel E and F

Referee #2 (Comments on Novelty/Model System for Author):

The method of developing a visual DEHP tracing system based on a Cy5-labeled DNA aptamer is innovative and useful, because it enables the real-time tracking of DEHP distribution and excretion in mouse primary hepatocytes and livers, which can facilitate the investigation of pharmacodynamics and toxicodynamics, as well as the screening of targeted drugs. However, there are also some challenges and uncertainties associated with this method, such as the selection and design of the DEHP aptamer, the validation of its affinity and specificity, the stability and biocompatibility of the Cy5 label, and the quantification and normalization of the fluorescence signals. Therefore, it is suggested that the authors should provide more details and explanations on how they developed and validated this method in the methods and results sections, and also perform some quality control and calibration experiments to ensure the reliability and accuracy of this method.

The technique of combining system and network pharmacology analyses to screen potential drugs from TCM prescriptions and chemical components is rational and effective, because it can enrich the sources and diversity of drug candidates, and also reveal the potential mechanisms and targets of the drugs. However, there are also some limitations and assumptions associated with this technique, such as the completeness and accuracy of the databases, the criteria and parameters of the screening and analysis, and the validation and verification of the results. Therefore, it is suggested that the authors should provide more information and justification on how they performed this technique in the methods and results sections, and also conduct some experimental and clinical studies to confirm and evaluate the efficacy and safety of the drugs.

The ethical issues that may arise from using mouse as the model organism are mainly related to the animal welfare and the potential harm caused by the exposure and treatment of DEHP, luteolin, trans-UCA, or AAV8-Uroc1. Therefore, it is suggested that the authors should follow the relevant guidelines and regulations for the care and use of laboratory animals, and obtain the approval and permission from the ethical committee or the institutional review board before conducting the animal experiments. The authors should also minimize the number and suffering of the animals, and use the appropriate doses and routes of administration of the substances. The authors should also report the ethical aspects of the animal experiments in the methods section, and disclose any potential conflicts of interest or competing interests in the paper.

Referee #2 (Remarks for Author):

This article studies the harm and challenge of DEHP as an environmental endocrine disruptor to human health, and the lack of effective high-throughput screening techniques and targeted drugs. It uses a self-developed visual DEHP tracking system and network pharmacology analysis method to screen and verify luteolin as a potential detoxification drug. The article expounds the mechanism of action of luteolin, that is, by targeting the Ala270 and Val272 sites of Uroc1, it promotes its protein ubiquitination degradation, thereby reducing its activity, increasing the level of its substrate urocanic acid, and then promoting the excretion of DEHP. This article reveals an endogenous detoxification metabolite, namely trans-UCA, which can simulate the detoxification effect of luteolin, inhibit the ERK1/2 signaling pathway, regulate lysosomal exocytosis, and promote the excretion of DEHP, which has good innovation and significance. I also have the following suggestions for modification and improvement: In the results of verifying the effect of luteolin and trans-UCA on DEHP excretion, the author did not explain how to measure the content of DEHP and MEHP in mouse urine, and how to control the collection and preservation conditions of mouse urine, to avoid the degradation or loss of DEHP and MEHP. It is suggested that the author explain the method and parameters of urine detection in the results section, such as the conditions of HPLC-MS, the selection of internal standards, the dilution and treatment of urine, etc. At the same time, it is suggested that the author explain the method and standard of urine collection and preservation in the method section, such as the collection time, collection method, collection amount, preservation temperature, preservation time, etc. In the results of label-free quantification and protein identification, the author did not explain how to screen and verify the candidate proteins regulated by luteolin or trans-UCA, and how to perform gene ontology (GO) analysis and pathway analysis. It is suggested that the author explain the criteria and methods of screening and verification in the results section, such as differential expression fold change, P value, false discovery rate, immunoblotting, immunofluorescence, etc. At the same time, it is suggested that the author display the results and charts of GO analysis and pathway analysis in the results section, such as enrichment analysis, functional classification, signal pathway, etc. In the discussion section, the author did not fully discuss the similarity and difference of the mechanism and effect of luteolin and trans-UCA on DEHP detoxification and liver protection, as well as their synergistic or antagonistic effect. It is suggested that the author further discuss the similarity and difference of the mechanism and effect of luteolin and trans-UCA on DEHP detoxification and liver protection, as well as their synergistic or antagonistic effect, in the discussion section, to improve the depth and breadth of the discussion. At the same time, it is suggested that the author cite some relevant literature in the discussion section, to support and compare the findings and conclusions of this study, to improve the logic and persuasiveness of the discussion.

Comments on the reviews

We extend our profound gratitude to the reviewers for their meticulous scrutiny of our submission. The insights they provided have indeed proven crucial in evaluating our research. We acknowledge and concur with all their noted concerns. In response, we have embarked on extensive investigations to address these issues, ensuring scientific rigor. Consequently, our manuscript has undergone substantial revisions and enhancement in depth. Below, we present a detailed point-by-point examination of each reviewer's points, accompanied by the strategies we employed to rectify and strengthen the respective aspects of our work.

Referee #1

1. Collateral organ damage, with a particular focus on the kidney: apart from the liver, the main target of phthalates is the kidney (see e.g. PMID: 38107267, 33736212, PMID: 34126474, PMID: 30462244, PMID: 24178978, PMID: 26100504). As reducing liver accumulation could lead to chronic over-exposure to DEHP/MEHP/MEHHP/MEOHP for the kidneys (PMID: 38107267), where they induce EMT and fibrosis (PMID: 29669060), and this could imply damage in terms of filtration rate or albuminuric loss, although luteolin in could be protective in this sense. As the authors aim to identify a potential anti-DEHP drug, they should also investigate the effect on the renal profile as described above (providing data on renal parameters and fibrosis at least).

Response: We appreciated the reviewer for her/his rigorous attitude in reviewing our manuscript. As requested, we firstly performed histological analyses to investigate the renal injury induced by DEHP. As shown in Fig. EV4F, H&E and Masson's trichrome staining analyses revealed that administration of DEHP led to significant renal impairment as evidenced by exacerbated tubular dilation, cast formation, loss of brush borders, and an increased presence of collagen fibers within the kidneys of DEHP-exposed mice. Notably, the renal expression levels of EMT biomarkers, including N-cadherin, Vimentin and α -SMA, exhibited substantial upregulation in response to DEHP exposure (Fig. EV4G). Analogous to its proven hepatoprotective effects, luteolin exhibited a renoprotective effect by mitigating DEHP-induced renal impairment and concomitantly decreasing the protein expression of EMT biomarkers within the kidney of DEHP-treated mice. These findings indicated the dual beneficial role of luteolin in protecting both hepatic and renal function of DEHP-treated mice. Nonetheless, the pivotal question that remains to be elucidated is the interdependence: whether the renal

protective effect of luteolin is intrinsically linked to its facilitation of hepatic DEHP detoxification or constitutes an independent protective mechanism. This inquiry is of paramount importance for uncovering the intricate molecular interactions and therapeutic implications of luteolin's targets.

2. In addition to the previous comment, the metabolic phenotype of the mice should at least be examined in summary: were there any changes in body weight, glucose and blood pressure levels between ctrl, DEHP and DEHP+luteolin mice? Was there an over-accumulation of DEHP in adipose tissue?

Response: As requested, we examine the metabolic phenotype of the mice. As shown in Fig. EV4A and EV4B, solitary exposure to DEHP, as well as its synergistic interaction with luteolin, did not yield any deviation from standard mouse body weight or blood glucose levels, maintaining them within normal reference ranges. The meticulous evaluation of blood pressure, through the analysis of serum angiotensin concentrations, further substantiated this stability, with the biomarker consistently present in all experimental cohorts (Fig. EV4C). Therefore, these findings implied that neither compound exerted a substantial impact on the metabolic profile of the test subjects. Conversely, luteolin demonstrated a pronounced effect in restraining the accumulation of DEHP and MEHP within adipose tissue (Fig. EV4D, E). Considering the distinctive protein expression of Uroc1 within the liver (<https://www.proteinatlas.org/ENSG00000159650-UROC1>), this reduction could be predominantly attributed to luteolin's potent systemic protective mechanisms when mitigating the hepatotoxic effects of DEHP.

3. The authors mention in the text several times the adoption of untargeted proteomics (see results "Hepatic Uroc1 is responsive to luteolin treatment at the translational level" and "ERK1/2-orchestrated lysosome pathway is involved in mediating the luteolin/trans-UCA-facilitated hepatic DEHP release" paragraphs, and the figures 3 and 6). Nevertheless, in the same figures, and in the correspondent text, it seems that the authors have rather adopted a transcriptomic screening (in the subsequent panels they refer to genes, not to proteins, e.g. "As shown in Fig. 3B, a cluster of 21 genes was changed in response to luteolin treatment"; "A total of 76 genes were altered by trans-UCA treatment, with 27 upregulated and 49 downregulated") which is not described in the methods (were they RNA-seq? custom/non-custom microarray screening?). Might

the author clarify on this? Are the authors willingly to deposit the omics data in a public repository?

Response: Thanks. The high-throughput analyses throughout our manuscript were proteomics, we have corrected all the improper statement, “As shown in Fig. 3B, a cluster of 21 proteins was changed in response to luteolin treatment”; “A total of 76 proteins were altered by *trans*-UCA treatment, with 27 upregulated and 49 downregulated”. The mass spectrometry proteomics data have been deposited to the ProteomeXchange Consortium (<https://www.ebi.ac.uk/pride/>) via the PRIDE partner repository with the dataset identifier PXD055165 for Luteolin (Reviewer account details: Username: reviewer_pxd055165@ebi.ac.uk, Password: 2cEOJJ3jNU9i) and PXD055187 for *Trans*-UCA (Reviewer account details: Username: reviewer_pxd055187@ebi.ac.uk, Password: ZMXcFQFVjilW).

4. The authors focused on intragastric absorption, and this should be better clarified in the manuscript (especially when discussing implications, limitations and discussion), as it is known that there are other major routes of absorption and that organ accumulation also depends on the route of absorption (e.g. PMID: 35565138), which slightly limits the generalisability of the authors' work in terms of actual treatment of DEHP exposure.

Response: This is an important concern. DEHP, with its persistence in the food chain, is commonly found in contaminated oil crops and animal fats, as well as in plastic packaging where it leaches into consumables during storage (Chen et al, 2012; Gao et al, 2023). Oral intake, especially from contaminated food, accounts for over 90% of DEHP absorption in humans, which underscores the significance of the liver as the primary site of metabolism for this compound (Li et al, 2022). Our study's focus on oral ingestion and the liver as the target, addressing the potential of luteolin and its derivative *trans*-UCA in reducing DEHP and MEHP accumulation within the liver. However, it's crucial to acknowledge that DEHP's entry into the body takes various forms, such as dermal absorption, intravenous administration, and inhalation, each with its own specific organ distribution patterns (Li et al., 2022). The organs, such as pancreas and spleen, particularly in cases of dermal penetration, exhibit higher concentrations. Therefore, future research investigating the detoxifying effects of

luteolin in these alternative sites would broaden its applicability in against DEHP exposure through different routes, making it a more versatile strategy for combating environmental pollutants. These studies will contribute to a more comprehensive and generalizable understanding of luteolin's protective role against DEHP toxicity.

5. Might the authors provide a GO analysis for luteolin modulated pathways in figure 3, as provided in figure 6 for *trans*-UCA to observe more easily the difference between the metabolite replenishment vs the drug approach? (e.g., *Acaa1b* has a opposite behaviour).

Response: As requested, we have performed a GO analysis for luteolin-modulated pathways in DEHP-treated mouse primary hepatocytes. As shown in Fig. EV5A and Appendix Table S6, GO analysis disclosed alterations in biological processes like cellular component synthesis and assembly, which is consistent with previous findings that highlighted luteolin's role in maintaining the cellular homeostasis of hepatocytes (Li et al, 2020). Although luteolin did not show the same Top15 pathways as *trans*-UCA (Appendix Table S8), the MAPK pathways (which was enriched in *trans*-UCA-treated group) were notably clustered, indicating an involvement in the luteolin-regulated network (Highlighted in Appendix Table S6). The discrepancy in the Top15 pathways could be attributed to the differential nature of luteolin and *trans*-UCA. Luteolin, being a small molecule, engages with a diverse range of targets due to its functional groups, while *trans*-UCA as an endogenous metabolite might yield more specific downstream pathways in mitigating DEHP accumulation.

In addition, the inverse changes in *Acaa1b* protein expression could result from distinct responses to the two compounds. Luteolin's possible effect on decreasing *Acaa1b* might stem from its lipids-lowering properties, which decreased the expression of fatty acid synthesis-associated enzymes, like FASN (decreased to 57.9% compared to DEHP-treated group), thus leading to a feedback mechanism to decrease *Acaa1b* protein expression. In contrast, *trans*-UCA's observed increase in *Acaa1b* could be related to enhanced fatty acid oxidation to fuel histidine metabolism and detoxification of DEHP. Considering that we focused on detoxifying effects of luteolin and *trans*-UCA, this inconsistent change of *Acaa1b* did not alter the net efficacy of either luteolin and *trans*-UCA, as evidenced by the MAPK pathways were enriched by GO analysis of proteomics results from these two compounds-treated mouse primary hepatocytes. Despite these inconsistencies, the enriched MAPK pathways in both luteolin and *trans*-

UCA treatments suggest that they maintain their overall efficacy in detoxifying processes.

6. Methods: the animal protocol states "These mice were treated with either DEHP or *trans*-UCA (10 mg/kg body weight/day) in a manner similar to the experiment evaluating the effect of luteolin." What is meant by "similar"? Was there any difference, however small? Could the author precisely report the study protocol for the *trans*-UCA?

Response: The reviewer is right. We have deleted the "similar" and clarified the *trans*-UCA treatment experiment in animal protocol. Please see the section of "Animals" in "Methods".

7. Statistical analysis: Was a correction for multiple testing correction (e.g. FDR) used for -omics results? This should be clarified in the main text (even if it was not used, it should be clearly reported in the manuscript).

Response: We have clarified this important information in the section of "Methods". The FDR correction serves as a vital statistical measure to minimize noise in the data, but it can lead to the elimination of potentially relevant findings (Lieberman & Cunningham, 2009), specifically when it comes to luteolin's molecular targets in DEHP-treated mouse primary hepatocytes. To overcome this limitation, we decided to expand our investigation by including all proteins that exhibited differential expression, without imposing the FDR filter. This broader approach allowed us to uncover novel interactions and targets, such as UROC1, which exhibited significant changes in response to luteolin treatment. The significance of UROC1 as a potential target was reinforced by experimental validation using Western blot technique, which directly confirmed the alterations in UROC1 protein levels observed through proteomics data. FDR correction can also lead to a reduction in true positives, restricting the GO analysis. Similar non-rectification techniques have also been employed elsewhere (Lieberman & Cunningham, 2009). As requested, we performed a FDR correction and did not observe significantly enriched pathways after adjustment. This might be attributed to the specific DEHP concentration used, which is designed to emulate everyday environmental exposure levels (Lieberman & Cunningham, 2009). Although this concentration induces a moderate molecular perturbation in hepatocytes, it is capable of demonstrating luteolin's function in mitigating accumulation of DEHP and its metabolite MEHP within the cells. Hence, we finally performed GO analysis according

to the dataset without FDR correction.

8. Figure 5: typo ("tran" instead of "*trans*"-UCA) in panel E and F.

Response: Already corrected.

Reference:

Chen L, Zhao Y, Li L, Chen B, Zhang Y (2012) Exposure assessment of phthalates in non-occupational populations in China. *Sci Total Environ* 427-428: 60-69

Gao X, Cui L, Mu Y, Li J, Zhang Z, Zhang H, Xing F, Duan L, Yang J (2023) Cumulative health risk in children and adolescents exposed to bis(2-ethylhexyl) phthalate (DEHP). *Environ Res* 237: 116865

Li A, Kang L, Li R, Wu S, Liu K, Wang X (2022) Modeling di (2-ethylhexyl) Phthalate (DEHP) and Its Metabolism in a Body's Organs and Tissues through Different Intake Pathways into Human Body. *Int J Environ Res Public Health* 19

Li T, Zhang L, Jin C, Xiong Y, Cheng YY, Chen K (2020) Pomegranate flower extract bidirectionally regulates the proliferation, differentiation and apoptosis of 3T3-L1 cells through regulation of PPARgamma expression mediated by PI3K-AKT signaling pathway. *Biomed Pharmacother* 131: 110769

Lieberman MD, Cunningham WA (2009) Type I and Type II error concerns in fMRI research: re-balancing the scale. *Soc Cogn Affect Neurosci* 4: 423-428

Referee #2

1. There are also some challenges and uncertainties associated with this method, such as the selection and design of the DEHP aptamer, the validation of its affinity and specificity, the stability and biocompatibility of the Cy5 label, and the quantification and normalization of the fluorescence signals. It is suggested that the authors should provide more details and explanations on how they developed and validated this method in the methods and results sections, and also perform some quality control and calibration experiments to ensure the reliability and accuracy of this method.

Response: This is an important issue. As requested, we have provided the detailed information in the "Methods" section and added essential parameters in the "Results" section. Please see Page 7, Line 185-195 and Page 21-24, Line 585-703. On the other hand, we agreed with reviewer in quality control and calibration experiments to ensure

the reliability and accuracy of our *csa*-DEHP system. To address this, we assessed a long-term stability of the *csa*-DEHP system by using *csa*-DEHP stored at -80°C for over 1 year. As shown in Appendix Figure S2A, although fluorescence intensity exhibited a slight decline over time, the overall stability of the system was maintained, as demonstrated by the consistent fluorescence signals retained for more than a year despite the cold storage conditions. The parallel test with ¹H NMR spectra provided additional validation (Appendix Figure S2B). No significant changes in the spectral characteristics, confirming that the chemical structure of DEHP was preserved without major degradation over the extended period. This result indicates that the stability of the *csa*-DEHP system at -80°C is suitable for maintaining the quality of its measurements and supports its application in long-term research projects and storage scenarios without compromising accuracy.

2. The technique of combining system and network pharmacology analyses to screen potential drugs from TCM prescriptions and chemical components is rational and effective, because it can enrich the sources and diversity of drug candidates, and also reveal the potential mechanisms and targets of the drugs. However, there are also some limitations and assumptions associated with this technique, such as the completeness and accuracy of the databases, the criteria and parameters of the screening and analysis, and the validation and verification of the results. Therefore, it is suggested that the authors should provide more information and justification on how they performed this technique in the methods and results sections, and also conduct some experimental and clinical studies to confirm and evaluate the efficacy and safety of the drugs.

Response: We appreciated the reviewer for her/his rigorous attitude in reviewing our manuscript. We have added indicated information of TCMSP including completeness and accuracy of the databases, the criteria and parameters of the screening and analysis, and the validation and verification of the results, as well as how we performed this technique in the sections of “Methods” and “Results”. Please see Page 20, Line 550-583 in the “Methods” section: “TCMSP serves as a premier resource for in-depth pharmacological studies on TCM, … … and candidate drugs decreasing DEHP accumulation were identified by screening for effective components with a degree of ≥ 3 ”. Page 5, Line 104-122 in the “Results” section: “To discover novel drugs for the detoxification of DEHP from the liver, we subjected a comprehensive examination to

128 detoxification formulas hailing from TCM.....As shown in Fig. 1B, a cluster of 37 monomers was identified, while 14 monomers with a degree number ≥ 3 were selected (Appendix Table S4) ”.

On the other hand, we totally agreed with the reviewer’s points that experimental and clinical studies were need to confirm and evaluate the efficacy and safety of the drugs. In our study, we evaluated the safety of screened drugs by using CCK-8 analysis in mouse primary hepatocytes, final focusing on 7 monomers. Furthermore, we assessed the efficacy of screened drugs by using *csa*-DEHP system and HPLC-MS analyses. We selected luteolin as a promising candidate due to its remarkable ability to significantly reduce DEHP levels in mouse primary hepatocytes. However, it is important to note that these findings provide early lines of evidence *in vivo*. Hence, we examined the detoxification efficacy and safety of luteolin. As shown in Fig. 2C and 2D, luteolin (10 mg/kg body weight/day) reduced excessive DEHP accumulation in mouse livers. These results suggested that luteolin indeed displays a verifiable capacity to effectively reduce the accumulation of DEHP. Furthermore, administration of luteolin did not yield any deviation from standard mouse body weight or blood glucose levels, maintaining them within normal reference ranges (Fig. EV4A-4B). The meticulous evaluation of blood pressure, through the analysis of serum angiotensin concentrations, further substantiated this stability, with the biomarker consistently present in all experimental cohorts (Fig. EV4C). Therefore, these findings implied that neither compound exerted a substantial impact on the metabolic profile of the test subjects. In addition, luteolin ameliorated DEHP-induced liver injury and macrophage infiltration, while serum AST and ALT levels were correspondingly decreased in mice, indicating the hepatoprotective effects of luteolin. Furthermore, reducing liver accumulation could lead to chronic over-exposure to DEHP/MEHP for the kidneys (Zhang et al, 2023), where they induce EMT and fibrosis (Wu et al, 2018), and this could imply damage in terms of filtration rate or albuminuric loss. Hence, to evaluate the potential collateral organ damage of DEHP, we performed histological analyses to investigate the renal injury induced by DEHP. As shown in Fig. EV4F, H&E and Masson-trichrome staining analyses revealed that administration of DEHP led to significant renal impairment as evidenced by exacerbated tubular dilation, cast formation, loss of brush borders, and an increased presence of collagen fibers within the kidney of DEHP-exposed mice. Notably, the renal expression levels of EMT

biomarkers, including N-cadherin, Vimentin and α -SMA, exhibited substantial upregulation in response to DEHP exposure. Analogous to its proven hepatoprotective effects, luteolin exhibited a renoprotective effect by mitigating DEHP-induced renal fibrosis and concomitantly decreasing the protein expression of EMT biomarkers within the kidney of DEHP-treated mice (Fig. EV4G). Clinically, luteolin formulation (NeuroProtek, which is approved by the U.S. Food and Drug Administration) containing 100 mg luteolin was found to inhibit mast cells and reduce brain inflammation in children with autism spectrum disorders (n=37; 4-14 years old), indicating the clinical safety of luteolin at the dose of 100 mg (Wang et al, 2021).

Collectively, luteolin, the compound systematically selected through the TCMSP platform and independently substantiated by both the *csa*-DEHP system and HPLC-MS, stands as a safe and efficacious agent, manifesting its inherent prowess in detoxifying DEHP, thereby exhibiting a promising profile for protective applications against environmental pollutants.

3. The authors should also report the ethical aspects of the animal experiments in the methods section, and disclose any potential conflicts of interest or competing interests in the paper.

Response: The reviewer is right. we have clarified this important information as requested. Please see Page 25, Line 727-732.

4. In the results of verifying the effect of luteolin and *trans*-UCA on DEHP excretion, the author did not explain how to measure the content of DEHP and MEHP in mouse urine, and how to control the collection and preservation conditions of mouse urine, to avoid the degradation or loss of DEHP and MEHP. It is suggested that the author explain the method and parameters of urine detection in the results section, such as the conditions of HPLC–MS, the selection of internal standards, the dilution and treatment of urine, etc. At the same time, it is suggested that the author explain the method and standard of urine collection and preservation in the method section, such as the collection time, collection method, collection amount, preservation temperature, preservation time, etc.

Response: This is an important suggestion. As requested, we have clarified this important information in the “Method” section. Please see Page 27-28 Line 784-821.

5. In the results of label-free quantification and protein identification, the author did not explain how to screen and verify the candidate proteins regulated by luteolin or *trans*-UCA, and how to perform gene ontology (GO) analysis and pathway analysis. It is suggested that the author explain the criteria and methods of screening and verification in the results section, such as differential expression fold change, P value, false discovery rate, immunoblotting, immunofluorescence, etc. At the same time, it is suggested that the author display the results and charts of GO analysis and pathway analysis in the results section, such as enrichment analysis, functional classification, signal pathway, etc.

Response: As requested, we have clarified this information and the results in the sections of “Methods” and “Results”. Please see Page 28-29, Line 832-845, and Page 9, Line 250-256, as well as Fig. EV5A and Appendix Table S6. While acknowledging the importance of comprehensive analysis in proteomics, we decided not to conduct KEGG pathway analysis for the following reasons: 1) The primary focus of our proteomics examination regarding luteolin treatment was to pinpoint the direct molecular targets of the compound. Given this objective, a more detailed KEGG pathway analysis might not be absolutely necessary, as the primary targets would be more relevant to understanding the mechanism of luteolin’s function. 2) For the proteomics study centered on *trans*-UCA, we deemed GO analysis of biological pathways sufficient to capture the downstream effects. In fact, MAPK pathway was also enriched in KEGG pathway analysis (Please see figures below). Please also refer to the answers of **Q6**, we also discussed the synergistic or antagonistic effects of luteolin and *trans*-UCA on DEHP detoxification and liver protection through the view of GO results.

Figures for review only.

6. It is suggested that the author further discuss the similarity and difference of the mechanism and effect of luteolin and *trans*-UCA on DEHP detoxification and liver protection, as well as their synergistic or antagonistic effect, in the discussion section, to improve the depth and breadth of the discussion. At the same time, it is suggested that the author cite some relevant literature in the discussion section, to support and compare the findings and conclusions of this study, to improve the logic and persuasiveness of the discussion.

Response: This is an awesome suggestion. In fact, we have performed a GO analysis for luteolin-modulated biological pathways in DEHP-treated mouse primary hepatocytes in our revised manuscript (as suggested by you and reviewer#1). As shown in Fig. EV5A, GO analysis disclosed alterations in biological processes like cellular component synthesis and assembly, which is consistent with previous findings that highlighted luteolin's role in maintaining the cellular homeostasis of hepatocytes(Li et al., 2020), Although luteolin did not show the same Top15 pathways as *trans*-UCA (Appendix Table S8), the MAPK pathways (which also enriched in *trans*-UCA-treated group) were notably enriched, indicating an involvement in the luteolin-regulated network (Highlighted in Appendix Table S6). The discrepancy in the Top15 pathways

could be attributed to the differential nature of luteolin and *trans*-UCA. Luteolin, being a small molecule, engages with a diverse range of targets due to its functional groups (Breger, 1970), while *trans*-UCA as an endogenous metabolite might yield more specific downstream pathways in mitigating DEHP accumulation.

In addition, the inverse changes in Acaa1b protein expression could result from distinct responses to the two compounds. Luteolin's possible effect on decreasing Acaa1b might stem from its lipids-lowering properties, which could promote the expression of other lipid β -oxidation enzymes, like FASN (decreased to 57.9% compared to DEHP-treated group), thus leading to a feedback mechanism to decrease Acaa1b protein expression. In contrast, *trans*-UCA's observed increase in Acaa1b could be related to enhanced fatty acid oxidation to fuel histidine metabolism and detoxification of DEHP. Considering that we focused on detoxifying effects of luteolin and *trans*-UCA, this inconsistent change of Acaa1b did not alter the net efficacy of either luteolin and *trans*-UCA, as evidenced by the MAPK pathways were enriched by GO analysis of proteomics results from these two compounds-treated mouse primary hepatocytes. Despite these inconsistencies, the enriched MAPK pathways in both luteolin and *trans*-UCA treatments suggest that they maintain their overall efficacy in detoxifying processes.

Reference:

- Wang Z, Zeng M, Wang Z, Qin F, Chen J, He Z (2021) Dietary Luteolin: A Narrative Review Focusing on Its Pharmacokinetic Properties and Effects on Glycolipid Metabolism. *J Agric Food Chem* 69: 1441-1454
- Wu CT, Wang CC, Huang LC, Liu SH, Chiang CK (2018) Plasticizer Di-(2-Ethylhexyl)Phthalate Induces Epithelial-to-Mesenchymal Transition and Renal Fibrosis In Vitro and In Vivo. *Toxicol Sci* 164: 363-374
- Breger I (1970) Affective response to meaningful sound stimuli. *Percept Mot Skills* 30: 842
- Chen L, Zhao Y, Li L, Chen B, Zhang Y (2012) Exposure assessment of phthalates in non-occupational populations in China. *Sci Total Environ* 427-428: 60-69
- Gao X, Cui L, Mu Y, Li J, Zhang Z, Zhang H, Xing F, Duan L, Yang J (2023) Cumulative health risk in children and adolescents exposed to bis(2-ethylhexyl) phthalate (DEHP). *Environ Res* 237: 116865
- Li A, Kang L, Li R, Wu S, Liu K, Wang X (2022) Modeling di (2-ethylhexyl) Phthalate (DEHP) and Its Metabolism in a Body's Organs and Tissues through Different Intake

Pathways into Human Body. *Int J Environ Res Public Health* 19

Li T, Zhang L, Jin C, Xiong Y, Cheng YY, Chen K (2020) Pomegranate flower extract bidirectionally regulates the proliferation, differentiation and apoptosis of 3T3-L1 cells through regulation of PPARgamma expression mediated by PI3K-AKT signaling pathway. *Biomed Pharmacother* 131: 110769

Lieberman MD, Cunningham WA (2009) Type I and Type II error concerns in fMRI research: re-balancing the scale. *Soc Cogn Affect Neurosci* 4: 423-428

Wang Z, Zeng M, Wang Z, Qin F, Chen J, He Z (2021) Dietary Luteolin: A Narrative Review Focusing on Its Pharmacokinetic Properties and Effects on Glycolipid Metabolism. *J Agric Food Chem* 69: 1441-1454

Wu CT, Wang CC, Huang LC, Liu SH, Chiang CK (2018) Plasticizer Di-(2-Ethylhexyl)Phthalate Induces Epithelial-to-Mesenchymal Transition and Renal Fibrosis In Vitro and In Vivo. *Toxicol Sci* 164: 363-374

Zhang Q, Qiu C, Jiang W, Feng P, Xue X, Bukhari I, Mi Y, Zheng P (2023) The impact of dioctyl phthalate exposure on multiple organ systems and gut microbiota in mice. *Heliyon* 9: e22677

1st Oct 2024

Dear Prof. Chen,

Thank you for the submission of your revised manuscript to EMBO Molecular Medicine. I am pleased to inform you that we will be able to accept your manuscript pending the following final amendments:

1) In the main manuscript file, please do the following:

- Please address all comments suggested by our data editors listed below:

o Data availability: Please note that the specific URLs for PXD055165 and PXD055187 datasets are not provided in the data availability statement.

o Figure legends:

1. Please define the annotated p values ****/###** as well as provide the exact p-values for the same in the legend of figure 2c-d; EV 5c; as appropriate.

2. Please note that the exact p values are not provided in the legends of figures 1e; 2e; 4a, f-h; 5c, e-g; 6d, f; EV 1d-e, h-j; EV 3j, l, n; EV 4d-e; EV 5d, f-h.

3. Please indicate the statistical test used for data analysis in the legends of figures 6c; EV 5a.

4. Please note that information related to n is missing in the legends of figures 2c-d; 3d-e; EV 1a-h; EV 3e; EV 5-c.

5. Although 'n' is provided, please describe the nature of entity for 'n' in the legends of figures 3i; 5c, e-g; EV 2e; EV 3g, j, l, n; EV 5d, f-g.

6. Please note that the scale bar is missing for figures EV 3f, h-i, k, m.

- Add callouts for panels A-L of Fig EV1.

- Author contributions: Please remove it from the manuscript and specify author contributions in our submission system. CRediT has replaced the traditional author contributions section because it offers a systematic machine-readable author contributions format that allows for more effective research assessment. Please use the free text boxes beneath each contributing author's name to add specific details on the author's contribution. More information is available in our guide to authors:

<https://www.embopress.org/page/journal/17574684/authorguide#authorshipguidelines>

- Thank you for including Reagent Tables. Please use the template you can download using the link below. More information on how to adhere to this format as well as downloadable templates (.docx) for the Reagents and Tools Table can be found in our author guidelines: <https://www.embopress.org/page/journal/17574684/authorguide#structuredmethods>

An example of a paper with Structured Methods can be found here:

<https://www.embopress.org/doi/full/10.1038/s44320-024-00037-6#sec-4>

- Please add description in the figure legend for Fig EV 4H.

- Indicate in legends number and nature of replicates and exact p= values, not a range, along with the statistical test used. To keep the figures "clear" some authors found providing an Appendix table Sx with all exact p-values preferable. You are welcome to do this if you want to.

- Rename Funding to Acknowledgements.

- In data availability section please add URLs for PXD055165 and PXD055187 datasets. Also remove the following sentences: "The source data of Figs. 1D, 1E, 2A-G, 3B-J, 4A, 4C-I, 5B-H, 6B, 6D-F, are provided as a Source Data file. All other data are available from the authors upon reasonable request."

-

Use the following format to report the accession number of your data:

[data type]: [full name of the resource] [accession number/identifier] [(doi or URL or identifiers.org/DATABASE:ACCESSION)]

Please check "Author Guidelines" for more information.

<https://www.embopress.org/page/journal/17574684/authorguide#availabilityofpublishedmaterial>

2) Tables: Please combine appendix tables S2, S4, S5, S7, S9, S10, S11, S12, S13, S14 and S15 in an Appendix and upload it as a single PDF file. Tables should be renumbered to Appendix Table S1 etc. and add table of content with page numbers on the title page of the Appendix file. Tables S1, S3, S6, S8 should be renamed to Table EV1 etc. Please also updated table callouts in the main manuscript text.

3) Source data: Remove checklist file from the zipped source data folders.

4) Synopsis:

- Synopsis image: Please simplify the synopsis image to increase readability and upload the image as a separate, high-resolution jpeg file 550 px-wide x (300-600) px-high.

5) As part of the EMBO Publications transparent editorial process initiative (see our Editorial at

<http://embomolmed.embopress.org/content/2/9/329>), EMBO Molecular Medicine will publish online a Review Process File (RPF) to accompany accepted manuscripts. This file will be published in conjunction with your paper and will include the anonymous referee reports, your point-by-point response and all pertinent correspondence relating to the manuscript. Let us know whether

you agree with the publication of the RPF and as here, if you want to remove or not any figures from it prior to publication. Please note that the Authors checklist will be published at the end of the RPF.

6) Please provide a point-by-point letter INCLUDING my comments as well as the reviewer's reports and your detailed responses (as Word file).

I look forward to reading a new revised version of your manuscript as soon as possible.

Yours sincerely,

Zeljko Durdevic

*** Instructions to submit your revised manuscript ***

To submit your manuscript, please follow this link:

<https://embomolmed.msubmit.net/cgi-bin/main.plex>

1) a .docx formatted version of the manuscript text (including Figure legends and tables)

2) Separate figure files*

3) supplemental information as Expanded View and/or Appendix. Please carefully check the authors guidelines for formatting Expanded view and Appendix figures and tables at <https://www.embopress.org/page/journal/17574684/authorguide#expandedview>

4) a letter INCLUDING the reviewer's reports and your detailed responses to their comments (as Word file).

5) The paper explained: EMBO Molecular Medicine articles are accompanied by a summary of the articles to emphasize the major findings in the paper and their medical implications for the non-specialist reader. Please provide a draft summary of your article highlighting

This may be edited to ensure that readers understand the significance and context of the research.

Please refer to any of our published articles for an example.

6) Author contributions: the contribution of every author must be detailed in a separate section.

7) EMBO Molecular Medicine now requires a complete author checklist (<https://www.embopress.org/page/journal/17574684/authorguide>) to be submitted with all revised manuscripts. Please use the checklist as guideline for the sort of information we need WITHIN the manuscript. The checklist should only be filled with page numbers where the information can be found. This is particularly important for animal reporting, antibody dilutions (missing) and exact values and n that should be indicated instead of a range.

8) Every published paper now includes a 'Synopsis' to further enhance discoverability. Synopses are displayed on the journal webpage and are freely accessible to all readers. They include a short stand first (maximum of 300 characters, including space) as well as 2-5 one sentence bullet points that summarise the paper. Please write the bullet points to summarise the key NEW findings. They should be designed to be complementary to the abstract - i.e. not repeat the same text. We encourage inclusion of key acronyms and quantitative information (maximum of 30 words / bullet point). Please use the passive voice. Please attach these in a separate file or send them by email, we will incorporate them accordingly.

You are also welcome to suggest a striking image or visual abstract to illustrate your article. If you do please provide a jpeg file 550 px-wide x 300-600px high.

9) A Conflict of Interest statement should be provided in the main text

10) Please note that we now mandate that all corresponding authors list an ORCID digital identifier. This takes <90 seconds to complete. We encourage all authors to supply an ORCID identifier, which will be linked to their name for unambiguous name identification.

Currently, our records indicate that the ORCID for your account is 0000-0002-3809-1062.

Link Not Available

11) Include a Reagents and Tools Table as part of the Methods section, which can be downloaded from our author guidelines (<https://www.embopress.org/page/journal/17574684/authorguide#structuredmethods>)

Photos 400-800 DPI

*Additional important information regarding figures and illustrations can be found at <https://bit.ly/EMBOPressFigurePreparationGuideline>. See also figure legend preparation guidelines: <https://www.embopress.org/page/journal/17574684/authorguide#figureformat>

***** Reviewer's comments *****

Referee #1 (Remarks for Author):

The revision made by the authors satisfies the previous comments of this referee. In my opinion, the manuscript is now methodologically clearer and more precise, and the investigation of the mechanisms is more comprehensive. I have no further comments.

Referee #2 (Remarks for Author):

Is suitable for publication.

Comments on the reviews

Thank you for the reviewers' comments concerning our manuscript entitled "Luteolin detoxifies DEHP and prevents liver injury by degrading Uroc1 protein in mice" (Manuscript ID: EMM-2024-19946-V3). These comments are all valuable and very helpful for improving our paper.

Referee #1 (Remarks for Author):

1. The revision made by the authors satisfies the previous comments of this referee. In my opinion, the manuscript is now methodologically clearer and more precise, and the investigation of the mechanisms is more comprehensive. I have no further comments.

Response: We thanked the reviewer for her/his awesome work in reviewing our manuscript.

Referee #2 (Remarks for Author):

1. Is suitable for publication.

Response: We thanked the reviewer for her/his awesome work in reviewing our manuscript.

11th Oct 2024

Dear Prof. Chen,

We are pleased to inform you that your manuscript is accepted for publication and is now being sent to our publisher to be included in the next available issue of EMBO Molecular Medicine.
